

# Impacts of Horizontal Resolution and Air–Sea Flux Parameterization on the Intensity and Structure of simulated Typhoon Haiyan (2013)

Mien-Tze Kueh[1], Wen-Mei Chen[1], Yang-Fan Sheng[1], Simon C. Lin[2], Tso-Ren Wu[3], Eric Yen[4], Yu-Lin Tsai[3], Chuan-Yao Lin[1]

[1]Research Center for Environmental Changes, Academia Sinica, Taipei, Taiwan
[2]Academia Sinica Grid Computing Centre, Institute of Physics, Academia Sinica, Taipei, Taiwan
[3]Institute of Hydrological and Oceanic Sciences, National Central University, Taiwan
[4]Academia Sinica Grid Computing Centre, Academia Sinica, Taipei, Taiwan

*Correspondence to*: Chuan-Yao Lin (yao435@rcec.sinica.edu.tw)

**Abstract.** This study investigates the impacts of horizontal resolution and surface flux formulas on typhoon intensity and structure simulations through the case study of the Super Typhoon Haiyan (2013). Three different sets of surface flux

formulas in the Weather Research and Forecasting Model were tested using grid spacing of 1, 3, and 6 km. Both increased resolution and more reasonable surface flux formulas can improve typhoon intensity simulation, but their impacts on storm structures are different. A combination of decrease in momentum transfer coefficient and increase in enthalpy transfer coefficients has greater potential to yield stronger storm. This positive effect of more reasonable surface flux formulas can be efficiently enhanced when the grid spacing is appropriately reduced to yield intense and contracted eyewall structure. As

resolution increases, the eyewall becomes more upright and contracted inward. The size of updraft cores in the eyewall shrinks and the region of downdraft increases; both updraft and downdraft become more intense. As a result, the enhanced convective cores within the eyewall are driven by more intense updrafts within a rather small fraction of spatial area. This contraction of eyewall is associated with an upper level warming process, which may be partly attributed to air detrained from the intense convective cores. This resolution dependence of spatial scale of updrafts is related to the model effective

resolution as determined by grid spacing.

## 1 Introduction

Intensity forecasting of tropical cyclones (TCs) remains a major challenge to numerical weather prediction. Numerous small-scale processes are involved in TC development, most of which are parameterized in numerical weather models. Therefore, the accuracy of physical representations in numerical models is crucial to TC intensity prediction (e.g., Bao et al., 2012). By

contrast, several studies have shown that increasing horizontal resolution has a positive effect on TC intensity and structure simulation (e.g., Davis et al., 2008; Fierro et al., 2009; Nolan et al., 2009; Gentry and Lackmann, 2010; Kanada and Wada,



2016). Of these, the majority have recognized that sufficiently high resolution is responsible for development of deep and intense eyewall updraft and thus a more organized inner-core structure of TCs (Fierro et al., 2009; Nolan et al., 2009; Gentry and Lackmann, 2010; Gopalakrishnan et al., 2011; Kanada and Wada, 2016). However, even with a very high model resolution, physical processes remain a source of uncertainty in numerical weather prediction of TC intensity (e.g., Fierro et

al., 2009; Gentry and Lackmann, 2010). For instance, the predictions of TC intensity and structure are sensitive to the model representation of physical processes involving planetary boundary layer evolutions (Braun and Tao, 2000; Hill and Lackmann, 2009; Bao et al., 2012; Nolan et al., 2009; Kanada et al., 2012; Coronel et al., 2016) and surface exchange fluxes (Braun and Tao, 2000; Davis et al., 2008; Hill and Lackmann, 2009; Bao et al., 2012; Nolan et al., 2009; Chen et al., 2010; Green and Zhang, 2013; Zhao et al., 2017).

Gopalakrishnan et al. (2011) noted that at higher resolution, larger moisture fluxes (resulting from strong convergences in the boundary layer) can lead to stronger storms. Kanada et al. (2012) indicated that large heat and water vapor transfers— essential for producing very intense TCs with an upright and contracted eyewall structure—can be attributed to the large vertical eddy diffusivities in lower boundary layers. Nolan et al. (2009) studied the sensitivity of TC inner-core structures to planetary boundary layer (PBL) parameterizations. They found that improvement of the representation of surface fluxes in

association with these PBL schemes could improve their respective simulation of TC boundary layer structure; however, discrepancies between the PBL schemes remained. By conducting 2-km resolution experiments, Coronel et al. (2016) found that both the surface drag and vertical mixing in the boundary layer were capable of substantially changing low-level wind structures. The authors further revealed that the mechanisms behind the two effects were different. Braun and Tao (2000) studied the sensitivity of TC intensity to PBL schemes and surface flux representations using a set of 4-km resolution

experiments. They concluded that representation of surface heat fluxes was the most determinative factor in TC intensity simulation. Bao et al. (2012) indicated that different PBL schemes lead to differences in the structure of simulated TC, whereas the near surface drag coefficient controls the relationship between the maximum wind speed and minimum central pressure.

The air-sea surface flux exchanges of moist enthalpy (via surface latent heat and sensible heat fluxes) acts as the primary

energy source, whereas the surface momentum flux (the surface drag) is the sink of the TC development. The potential intensity for a steady-state (or mature) tropical cyclone has long been hypothesized to be proportional to the ratio of surface bulk transfer coefficients for enthalpy and momentum, $C_K/C_D$ (Emanuel, 1986, 1995). The importance of surface flux formulas to TC intensity and structure simulations has also been recognized by numerous studies (e.g., Braun and Tao 2000; Davis et al., 2008; Hill and Lackmann, 2009; Bao et al., 2012; Nolan et al., 2009; Chen et al., 2010; Green and Zhang, 2013).

In recent years, observational estimates of surface fluxes revealed that the momentum transfer coefficient exhibits a steady increase to wind speed of approximately 30 $m\,s^{-1}$ but then levels off for higher wind speeds (e.g., Powell et al., 2003; Jarosz et al., 2007; French et al., 2007). For the extremely high wind speeds (e.g., exceeds 50 $m\,s^{-1}$), observational estimates for the momentum transfer coefficient are rather scattered (Bell et al., 2012; Richter et al., 2016). Both near surface





wind speed and surface sea state are the contributing factors in the formulation of bulk transfer coefficient for surface fluxes (e.g., Geernaert et al., 1987; Smith, 1988; Fairall et al., 2003; Zweers et al., 2010). Some studies have introduced the wind wave information into the momentum flux parameterization (e.g., Fairall et al., 2003; Zweers et al., 2010; Zhao et al., 2017). Zweers at al. (2010) shown that their sea spray dependent surface drag coefficient is in good agreement with observational

estimates reported by Powell et al. (2003), and leads to improvement in TC intensity forecasts. However, there is still a large degree of uncertainty in the estimates of these parameters. For example, Green and Zhang (2013) performed the intensity simulation for Hurricane Katrina (2005) using wind dependent fluxes exchange coefficients and all their experiments overestimated the storm intensity. They attributed this to the lack of ocean cooling effect in their atmosphere-alone model simulations. Zhao et al. (2017) considered three physical processes, namely breaking wave-induced sea spray, non-breaking-

induced vertical mixing, and rainfall-induced sea surface cooling, using a fully coupled atmosphere-ocean-wave model to improve the tropical cyclone intensity simulation. As compared with the observations, their simulations produced a comparable intensity for the Super Typhoon Haiyan (2013), but severely overestimated the intensity of another weaker typhoon. It appears that there is still a lack of a surface fluxes parameterization appropriate for the full spectrum of TC intensity. There is also need to have further study of the inter-relationship between surface fluxes and other model

configuration, including cloud microphysics as well as grid spacing. For example, the effect of grid spacing on surface flux parameterization regarding TC structure and intensity simulation has not yet been explored.

In this paper, we investigate the impacts of model horizontal resolution and surface flux formulas on TC structure and intensity simulations. We intend to study the sensitivity of TC intensity simulations to surface fluxes parameterizations when model resolution approaches the convective scale. The sensitivity experiments are performed on Super Typhoon Haiyan

(2013) with various resolutions and surface flux formulas. Typhoon Haiyan, the strongest TC in 2013, was a category 5-equivalent super typhoon on the Saffir–Simpson hurricane wind scale. Because of its minimum central pressure of 895 hPa, Haiyan became the strongest TC at landfall to date in the western North Pacific Ocean (Schiermeier, 2013; Lander et al., 2014). Lin et al. (2014) suggested that Haiyan's intensity can be considered "a league" higher than the majority of existing category 5 typhoons on the Saffir–Simpson wind scale. Such super typhoons are expected to increase both in number and

intensity under climate change (e.g., Bender et al., 2010; Schiermeier, 2010; Kanada et al., 2012; Lin et al., 2014), which brings even more challenges to TC prediction. The Super Typhoon Haiyan (2013) moved west-northwestward over the warm ocean and reaches its central pressure of 895 hPa several hours before it made its first landfall. The non-recurved track tendency allows us to focus on the intensity simulation. Section 2 describes the experimental setup, including the description of the surface flux formulas and experimental designs. Sections 3 and 4 present the results in terms of temporal evolution

and radius–height structure, respectively. Section 5 provides statistical and energy spectral analyses, followed by conclusions in Section 6.




## 2 Flux parameterization and experimental designs

In this study, the Weather Research and Forecasting (WRF)—The Advanced Research WRF modeling system (version 3.8.1; hereafter, WRF-ARW) was used (Skamarock et al., 2008). In all simulations, all physics options other than the surface flux options (described below) were identical, including the WRF single-moment three-class microphysics scheme (Hong et al.,

2004), updated Kain–Fritsch scheme (Kain and Fritsch, 1990; Kain, 2004) with a moisture-advection-based trigger function (Ma and Tan, 2009), Rapid Radiation Transfer Model for Global Circulation Models (RRTMG) shortwave and longwave schemes (Iacono et al., 2008), Yonsei University boundary layer scheme (Hong et al., 2006) used with the revised MM5 similarity surface layer scheme (Jimenez et al., 2012), and the unified Noah land surface model (Tewari et al., 2004) over land. In WRF, the bulk transfer coefficients as well as all surface fluxes (momentum, sensible heat, and latent heat) are

parameterized in a surface layer scheme. A brief description of the surface flux parameterization in WRF is provided in Appendix A for reference.

The behaviors of bulk transfer coefficients are strongly dependent on the momentum and/or scalar (heat and moisture) roughness lengths [cf. Eqs (A7), (A8), and (A9)]. Over the ocean, roughness length depends on the varying surface sea state which, to a large degree, ascribed by wind waves (e.g., Geernaert et al., 1987; Smith, 1988; Fairall et al., 2003). Under high

wind conditions, wave breaking and wind tearing the breaking wave chest generate sea foam and sea spray, thereby leading to significant regime change in the surface sea state which in turn effect the specification of the roughness length (e.g., Powell et al., 2003; Donelan et al., 2004; Zweers et al., 2010). For the surface layer physics over ocean, WRF-ARW provides alternative formulations aerodynamic roughness lengths of the surface momentum and scalar fields for tropical storm application, which can be set through the namelist variable isftcflx. There are three isftcflx options available in the

version 3.8.1 of WRF-ARW. Herein, we study the resolution dependence of all three. Each flux option is referred to by its corresponding number (0, 1, and 2) in the WRF namelist file.

Below, we describe the three sets of formulas of roughness lengths for momentum, sensible heat, and latent heat that have been implemented in the revised MM5 scheme in WRF version 3.8.1. For the sake of clarity, we calculated the bulk transfer coefficients and the momentum scaling parameter (friction velocity) with respect to the three sets of implemented roughness

length formulas in the case of neutral stability and a reference height of 10 m, using a set of pseudo wind speeds. Under neutral condition, the stability parameters $(z + z_0)/L$ and $(z + z_h)/L$ are zero [cf. Eqs (A7), (A8), and (A9)], therefore all terms that involved stability function can be dropped. The results are depicted in Figs. 1 and 2.

For the default flux option (F0), the momentum roughness length is given as Charnock's (1955) expression plus a viscous term, following Smith (1988):

$$z_0 = \alpha \frac{u_*^2}{g} + \frac{0.11v}{u_*} \qquad (1)$$

where $\alpha$ is the Charnock coefficient; $v$ is the kinematic viscosity of dry air, a constant value of $1.5 \times 10^{-5} \ m^2 s^{-1}$ is used. The Charnock's expression relates the aerodynamic roughness length to friction velocity for describing the gross effect





of wavy sea surface which is generated and supported by wind stress; the viscous term describes roughness behavior under smooth flow condition. The use of this roughness length formula yields a monotonic increase in the momentum transfer coefficient, as well as roughness length, with wind speed exceeds ~3.0 $\text{m } s^{-1}$ (Fig. 1; black thick solid curve). A constant value of $\alpha = 0.0185$ is used for F0.

The moisture roughness length is adapted from COARE 3.0 (Fairall et al., 2003), expressed as a function of the roughness Reynolds number, with an addition lower limit of $2.0 \times 10^{-9}$ use in this option:

$$z_q = max[\, 2.0 \times 10^{-9}, min(1.0 \times 10^{-4}, 5.5 \times 10^{-5} R_*^{-0.6})] \qquad (2)$$

where $R_* = z_0 u_*/v$ is the roughness Reynolds number; $v$ is the kinematic viscosity of dry air and is a function of air temperature for the $R_*$ calculation. The temperature roughness length $z_h$ is set equal to $z_q$. The upper limit of $1.0 \times 10^{-4}$

acts to keep the scalar roughness length invariant at very low wind speeds (below ~4.0 $\text{m } s^{-1}$), whereas the lower limit of $2.0 \times 10^{-9}$ is implemented to prevent the model from blowing up. Below this high wind speed, the heat/moisture transfer coefficient increases monotonically with wind speed (Fig. 1; gray thin solid and black dashed curves). Note that for this option, the formulas of roughness lengths are different from that used by Green and Zhang (2013), where the viscous term for $z_0$ was given as a constant value of $1.59 \times 10^{-5}$, $z_h$ is set equal to $z_0$, and $z_q$ is set to a somewhat different

functional form of $z_0$ and $u_*$.

For the flux option 1 (F1), the momentum roughness length is expressed as a blend of two roughness length formulas (Green and Zhang, 2013):

$$z_0 = max[\, 1.27 \times 10^{-7}, min(z_w z_2 + (1 - z_w)z_1, 2.85 \times 10^{-3})\,] \qquad (3a)$$

$$z_w = min\left(1, \left[\frac{u_*}{1.06}\right]^{0.3}\right) \qquad (3b)$$

$$z_1 = 0.011 \frac{u_*^2}{g} + 1.59 \times 10^{-5} \qquad (3c)$$

$$z_2 = \frac{10}{exp\left(9.5 u_*^{-1/3}\right)} + \frac{0.11v}{max(u_*, 0.01)} \qquad (3d)$$

where the first roughness length formula ($z_1$) is again the Charnock's (1955) expression plus a constant value of viscous term, with $\alpha = 0.011$ as suggested by Smith (1988). The second formula ($z_2$) is the exponential expression from Davis et al (2008) plus a viscous term, here a constant kinematic viscosity $v = 1.5 \times 10^{-5} \ m^2 s^{-1}$ is used. The two roughness

length formulas are combined by using a weight function ($z_w$), with a lower and upper limit on $z_0$ of $1.27 \times 10^{-7} m$ and $2.85 \times 10^{-3} m$, respectively. The lower and upper limits on $z_0$ are adapted from Davis et al (2008), with a slight different value of the lower limit. The upper limit used here acts to prevent a monotonic increase in the momentum roughness length,





as well as the momentum transfer coefficient, under high wind conditions. The resulting curves present an increase with wind speed below 33.0 m $s^{-1}$, whereas exhibit a leveling-off behavior starting at the wind speed of 33.0 m $s^{-1}$ (Fig. 1; red thick solid curves). The wind speed of 33 m $s^{-1}$ is very close to the lower limit of maximum wind speed for a typhoon. The use of this $z_0$ formula results in apparently smaller momentum roughness length and transfer coefficient at all wind

speeds than that in F0. This level-off in $C_D$ with wind speed suggests a decline in the efficiency of the exchange of momentum across the air-sea interface [cf. Eq. (A1)]. Considering that drag effect acts as a momentum sink in the surface layer, it is expected that an atmospheric simulation using the $z_0$ formula of F1 shall retain a larger portion of the total amount of near-surface momentum at all wind speeds than that using the F0 formula, and such difference in the transfer efficiency will become more and more significant with higher wind speeds because of the leveling-off behavior seen in F1.

F1 sets $z_h = z_q = 10^{-4}m$ for all wind speeds. Large and Pond (1982) suggested constant values for $z_h$ and $z_q$ under different conditions of stability, ranging from $10^{-9}m$ (stable) to $10^{-4}m$ (unstable). The constant value $10^{-4}m$ used here corresponds roughly to the unstable condition in Large and Pond (1982). While using an invariant scalar roughness length, the heat/moisture transfer coefficient can still vary with wind speed (Fig. 1; pink thin solid and red dashed curves) because of the drag-dependent effect [cf. Eqs. (A8) and (A9)]. As a result, the heat and moisture transfer coefficients are also

beginning to level off at the wind speed of 33.0 m $s^{-1}$.

For the flux option 2 (F2), the momentum roughness length is the same as that for F1. The temperature and moisture roughness lengths are expressed based on the formula proposed by Brutsaert (1975):

$$z_h = z_0 \, exp\big[-\kappa\big(7.3R_*^{1/4}P_r^{1/2} - 5\big)\big] \qquad (4)$$

$$z_q = z_0 \, exp\big[-\kappa\big(7.3R_*^{1/4}S_c^{1/2} - 5\big)\big] \qquad (5)$$

where $R_*$ is the roughness Reynolds number; $\kappa$ is the von Karman constant; $P_r$ and $S_c$ are the Prandtl and Schmidt numbers; 7.3 and 5 are experimental constant (Brutsaert, 1975b). As in F0, a temperature-depended $\nu$ is also used for $R_*$ calculation. The values of $\kappa$, $P_r$, and $S_c$ are set to 0.40, 0.71, and 0.60, respectively, as suggested by Brutsaert (1979). This set of formulas relates the heat/moisture roughness length to a scaling of $z_0$ by exponential expression, where the exponent contains information about the near surface flow regime as characterized by the roughness Reynolds number and

Prandtl/Schmidt numbers. The resulting $z_h$ and $z_q$ is practically an exponential decay of $z_0$ with wind speed (Fig. 1; green thin solid and green dashed curves). The scalar roughness lengths in F0 and F2 both decrease with wind speed, and their decreasing rates as well as magnitudes are similar at higher wind speeds. However, their resulting scalar transfer coefficients are apparently different. Unlike the monotonic increasing behavior found in F0, for F2 the heat and moisture transfer coefficients show upward trend with wind speed below 33.0 m $s^{-1}$ and then start decreasing at this wind speed. Such a


discrepancy between F2 and F0 results from the drag-dependent effect in the scalar transfer coefficients [cf. Eqs. (A8) and (A9)].

Figure 2 shows the bulk transfer coefficients and friction velocity obtained from existing observational estimates as found in the literatures. A brief description of these observational estimates is provided in Appendix B. Overall, the behavior of $C_D$

in F1 and F2 as a function of wind speed is more consistent with recent observations than that in F0 (Fig. 2 a). Under the low to moderate wind conditions, all the flux options generally predict similar behaviors as observed in the observational estimates, and their values are within the observed data range. For the higher wind conditions, F1, as well as F2, predict a level-off in $C_D$ values with wind speed such that it is significantly differ from F0 starting at 33 $\mathrm{m\ s^{-1}}$. Considering the large discrepancy as well as the large spread of data found in the observational estimates that beyond the wind speeds of

~55 $\mathrm{m\ s^{-1}}$, the leveling-off behavior of $C_D$ values in F1 (and thus F2) is more reasonable than the monotonic increasing $C_D$ values in F0 (Fig. 2a). This is also supported by the comparison of the variations in friction velocity as shown in Fig. 2b.

In Fig. 2 c, we plotted all observational estimates of scalar transfer coefficients ($C_H$, $C_Q$, and $C_K$) upon the availability in the literatures that we referred to for this study (as listed in Table B1). Visually, we see an upward trend between the data samples from Geernaert et al. (1987) and French et al. (2007), and no significant trend between French et al. (2007) and the

rest of two with higher wind conditions (Bell et al., 2012; Richter et al., 2016). However, Geernaert et al. (1987) have reported that their data were mostly collected under stable conditions. Previous studies have revealed a stability dependence of $C_H$, upon which values of $C_H$ under stable conditions are lower than that under unstable conditions (Large and Pond, 1982; Smith, 1988). According, the applicability of the increases in $C_H$ and $C_Q$ with wind speed remains elusive. As compare with F0, both F1 and F2 give relatively reasonable pattern of $C_H$ and $C_Q$, with F2 fits the observational estimates well than F1

does.

In the context of the leveling-off behavior of $C_D$ at high wind speed, as well as the absence of a monotonic increase in $C_H$ and $C_Q$ with wind speed, both F1 and F2 provide more reasonable formulas for parameterizing the surface fluxes than F0 does. Furthermore, F1 predicts larger $C_H$ and $C_Q$ at all wind speeds than F2 does, implying that F1 has a potentiality to gain larger enthalpy fluxes. Because the formula of $z_0$ (and thus $C_D$) is identical in F1 and F2 whereas their respective formulas

for $z_h$ and $z_q$ are different, and because the behaviors of transfer coefficients in F0 are quite distinct from those in F1 and F2, we use the ratio of $C_K/C_D$ as a metric for a comparison between the three flux options. By assuming that $C_H{\sim}C_Q{\sim}C_K$ here, we plot the ratios $C_H/C_D$ and $C_Q/C_D$ in Fig. 2 d. All the flux options present a decrease in the ratio with wind speed, except that for F1 an upward trend and level-off pattern are found under very low wind and high wind conditions, respectively. The $C_K/C_D$ derived from the available observational estimates were also imposed for reference. We noted that for moderate to

high wind speeds, the values of $C_K/C_D$ are all below the lower limit, namely 0.75, for mature hurricanes as suggested by Emanuel (1995). Even for F1, the ratio level-off at a value of 0.7. Relatively small value of $C_K/C_D$, as compared with the lower limit of 0.75, is indeed not uncommon and has been reported by numerous previous studies based on observational



measurements (e.g., Drennan et al., 2007; Zhang et al., 2008; Bell et al. 2012) and numerical simulations (e.g., Hill and Lackmann, 2009; Green and Zhang, 2013). Nonetheless, we still can qualitatively relate the tropical cyclone intensity to the ratio of $C_K/C_D$. Accordingly, F1 is expected to have highest potential to achieve the most intense storm between the three options because the highest values of $C_K/C_D$ almost at all wind speeds. F0 has the lowest values of $C_K/C_D$ owing to its largest values of $C_D$, thereby having the lowest potential to produce comparable storm intensity as in F1.

Table 1 gives a summary of the experiments designed for this study. There were three resolution groups, with horizontal grid spacing of 1, 3, and 6 km. All three isftcflx surface flux options were tested using resolutions of 3 and 6 km. Only F0 and F1 were tested in the 1-km group. The CPU time increased more than 20-fold from 3 km to 1 km. Because the simulation result of F2 is somewhat between those of F0 and F1 for other resolutions, we omitted the F2 test at 1-km resolution. We considered 3kmF0 as the control experiment. All experiments used a single domain and 45 vertical levels. The model domain covered a spatial region bounded by 0°N–19.62°N latitudes and 117.88°E–147.12°E longitudes (Fig. 3; red box). The experiment procedure included a nudging and a sensitivity stage. For the nudging stage, the grid nudging (Staufer and Seaman, 1994) implemented in WRF was applied to each resolution group using the default flux option (F0). The numerical solutions were nudging toward the gridded reanalysis every 6 hours at all levels above the PBL for 24 hours starting at 0000UTC 4 November 2013 (Fig. 3; tan dots). This nudging setting enables the development of mesoscale processes near the surface. The National Centers for Environmental Prediction Global Forecast System operational global analysis dataset was used to provide the initial and boundary conditions as well as data for the grid nudging. The model sea surface temperatures (SSTs) were taken from the RTG-SST data set (Gemmill et al., 2007), with daily time resolution; there was no ocean feedback in this study. The SSTs varied during the nudging stage up until 0000 UTC 5 November 2013 and thereafter remained fixed throughout the sensitivity simulation such that the differences in underlying SSTs between the flux options became negligible. Consequently, sensitivity of typhoon intensity and structure to the flux options (thus the surface flux formulas) can be determined to a large extent. All the sensitivity experiments started at 0000 UTC 5 November 2013 and ended at 0600 UTC 8 November 2013 (Fig. 3; red dots). All results are from within the 78 hours that the experiments ran.

Typhoon Haiyan formed in the western North Pacific on 3 November 2013 and underwent a rapid intensification a few days later. Haiyan reached peak intensity of 895 hPa before it made a cataclysmic landfall near Guiuan, Eastern Samar, Philippines on 8 November 2013 at about 0440 PHT (UTC + 8). The storm then crossed the Philippines, migrated west-northwestward and made its final landfall in Vietnam. Our simulation period for the sensitivity experiments covers the stages of Haiyan's peak intensity and landfall. We focus on the mature stage of Haiyan before its landfall near Guiuan, Eastern Samar.

## 3 Temporal evolutions of track and intensity

All simulations show a similar west-northwestward movement toward land and appropriately follow the observed best track, but with slower translation speeds (Fig. 4a). The typhoon landfall at Samar Island is delayed between 3 and 5 hours in these





sensitivity simulations. Because of the delayed landfall, we compare the evolution of the typhoon related to its longitudinal position (Fig. 4b). In general, the simulations continue to intensify and reach their peak right after 1800 UTC 7 November 2013 and before landfall. We refer to the timespan, from 1800 UTC 7 November 2013 to 0000 UTC 8 November 2013, as the mature stage of the simulated typhoon. For each resolution group, F1 is the most intense in terms of minimum central

pressure, whereas the intensity of F2 is somewhat between that of F0 and of F1. The intensity is not very sensitive to the resolutions of 3 and 6 km, but it significantly increases as grid spacing is reduced to 1 km. The experiment 3kmF1 yields an intensity level comparable with that of 1kmF0, demonstrating the benefit of using a more reasonable flux option in improving intensity simulation. Accordingly, the combination of decrease in momentum transfer coefficient (generally tends to reduce the energy loss) and increase in enthalpy transfer coefficients (thus more energy gain) has greater potential to yield

stronger storm. Finally, higher resolution is more conducive to the intensification due to change in flux formulas.

The comparison between the simulated maximum winds near typhoon center and the best-track wind information is somewhat complicated owing to the wind speed averaging period (Kueh, 2012; Knapp et al., 2013). We use the International Best Track Archive for Climate Stewardship (IBTrACS) dataset for comparison, which compiles best-track information from various agencies worldwide (Knapp et al., 2010). The best-track information in Figs. 3 and 4a is taken from the WMO

subset of the IBTrACS (IBTrACS-WMO, v03r09). The WMO best track for Haiyan is taken from the best-track data provided by the Japan Meteorological Agency, which relies on a wind pressure relationship (WPR) where maximum wind is given in terms of 10-minute sustained winds (Koba et al., 1991). However, simulated maximum winds near typhoon center were derived from WRF hourly output instantaneous maximum 10-m wind speeds at the beginning of each hour. Figure 4c presents the scatterplot of maximum wind speed versus minimum central pressure. The discrepancy found between the

simulations and the best-track data may be partially due to different wind speed averaging periods. The WMO best-track wind pressure distribution appropriately follows the operational WPR, namely the reference curve Koba. We use a factor of 0.88, which is used operationally at the Joint Typhoon Warning Center (JTWC), to convert the 10-minute sustained wind in Koba's WPR to an equivalent 1-minute sustained wind. The modified Koba curve Koba_1min now agrees with the operational WPR used at the JTWC. The JTWC's WPR uses 1-minute sustained winds (Atkinson and Holliday, 1977; Knaff

and Zehr, 2007). In practice, operational PWRs were derived statistically to best fit a set of observed parameters (pressures and winds) by using the gradient-wind approximation as a fundamental guide. In the context of statistical regression and use of observed parameters, other information, such as the vortex size, ambient conditions, and geographical location, would be incorporated into these reference curves (Knaff and Zehr, 2007). To a certain degree, the operational WPR may be considered the best fit to the actual mass–wind relation of TCs. The winds and pressures of Haiyan from the JTWC best-

track data have been plotted upon Fig. 4c and the distribution well agree with the JTWC operational WPR, we thereby omitted them from the plot then. All our simulated wind pressure distributions present similar slopes as the two reference curves of 1-minute averaged winds, indicating that these simulations are valid for further investigation of the typhoon structures.





Bao et al (2012) have reported that the simulated wind pressure relationship would be controlled by the formulation of drag coefficient. From a set of simulations for North Atlantic TCs between 2008 and 2011, Green and Zhang (2013) found that different $C_D$ formulas affect wind pressure distribution (see Fig. 9 in Green and Zhang, 2013). We found insufficient evidence supporting any apparent slope change in wind pressure distributions of different flux options. This discrepancy may

be partially due to the use of the square of maximum 10-m wind speed in their wind pressure distribution plot. By comparing the respective linear regression curves between all experiments, we did find that both the differences in flux options and resolution result in slight change in the slope (figures not shown). However, no distinct dependence of the slopes on the choice of flux option and resolution could be identified. In our study, the differences in wind pressure distribution between these sensitivity simulations are demonstrated by their magnitudes. For each resolution group, F1 is the most intense in terms

of both minimum central pressures and maximum winds, whereas intensity of F2 is somewhat between that of F0 and F1. A comparison of resolution groups used in F0 reveals a significant increase in intensity at higher resolutions during the mature stage (where the storm is at higher strength), with 1kmF0 receiving larger increments. The set of experiments using F1 presents a similar pattern, with the larger increments in 1kmF1 being much more marked.

   In brief, both increased resolution and more reasonable flux parameterization (e.g., F1 in this case) can improve typhoon

intensity simulation to a certain extent. However, sufficiently high resolution is more conducive to the benefit from improved flux formulas. The implication is twofold. First, the use of reasonable flux formulas to improve intensity simulation is perhaps more efficient than using a very high resolution when computational resources are limited. Second, higher model resolution is conducive to improving surface flux representation in strong typhoon intensity simulations. The mechanisms underlying these two issues are notable. We address the issue of higher model resolution and its effect to

surface flux representation in the following sections.

**4 Storm structure at the mature stage**

   The sensitivity of simulated typhoon intensity to various resolutions and flux options is also associated with changes in the simulated typhoon structure. To isolate differences between simulations of different grid spacing, the hourly outputs on the WRF native grid were transformed to the cylindrical (polar-height) coordinate system centered at the simulated typhoon at

each time point. To avoid spatial sampling issues in the comparison of mass and wind fields, all model outputs were transformed to an identical resolution setting. We used a 3-km horizontal resolution and an uneven vertical resolution with smaller spacing in the boundary layer for the cylindrical transformation. Specifically, the cylindrical coordinate used in this study has an azimuthal resolution of 2 degree, a radial resolution of 3 km, and an uneven vertical resolution within the radius and height of 900 km and 18 km, respectively. All the data to be analyzed are interpolated from the model native coordinate

to the cylindrical coordinate using bilinear interpolation. The zonal and meridional components of horizontal wind fields are transformed to tangential and radial wind components using their standard formulas. The (height-invariant) typhoon center is defined as minimum central pressure using the surface pressure field. We also examined the geometric center for the local

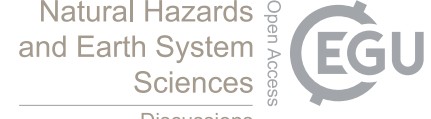



minimum pressure and found that the distances between the center at grid point and geometric center are less that the half of the grid spacing, for the simulation period after 24 h (valid at 0000 UTC 6 November 2013). The magnitudes of the vertical tilt of center (in term of the WRF output perturbation pressure field) are also small, about the size of one grid box. In this study, radial variations of model fields were all derived from these transformed subsets.

Figure 5 shows the horizontal distributions of wind speed at 10-m height, total rainfall, and surface enthalpy fluxes for F0 with 3-km resolution after the cylindrical transformation. The wind speed distribution shows a ring of higher wind speed (exceeds ~42 m $s^{-1}$), with wider regions of high wind speed over the right side (northern semicircle) of the track, with a partial ring of the maximum wind speed (~50 m $s^{-1}$), open to the west, at the range of radii 40-50 km. A region of intense rainfall is located on the left-rear side of the track. Regions of larger amount of surface fluxes are located nearby the local

maximum wind speeds; however, they are not necessarily co-located. This is because the near surface enthalpy vertical gradient, other than wind speed, is also an influential component in flux calculation. The discrepancies in the horizontal distributions of surface fields between different resolution groups are more significant than that between flux options (Fig. 6). The 1-km resolution group yields a significant eyewall (in term of these surface fields) contraction as compared with the other two groups. The 3-km resolution group exhibits a relatively larger eye region, less contracted radius of the maximum

wind speed, as well as total rainfall and surface fluxes. The 6-km resolution group does not present well-defined eyewall structure in term of these surface fields. Therefore, the increase in resolution from 6-km to 3-km yields a better representation of the concentric pattern of the near surface fields related to the typhoon center. In general, the use of different flux options result in the enhanced of these surface fields, and thus the intensity in term of the minimum central pressure and maximum wind speed, in the order of F1 > F2 > F0.

We have examined these surface fields over the period from 1200UTC 7 November to 0000UTC 8 November and found similar features as in Fig. 6, indicating that this period can be seen as the mature stage of the simulated typhoon. Overall, the horizontal distributions of the wind speed at 10-m height and total rainfall reveal a region of higher wind speed on the right (northern) side of the track, and a region of intense rainfall to the left side of rear side of the track. This result consists with the study of Shimada et al. (2018) for Haiyan Typhoon. Based on radar reflectivity measured several hours before the first

landfall of Haiyan, they found that region of the larger maximum wind speed was located on the right side of the track, and the region of stronger retrieved rainfall was located downshear-left side. During their analysis period (about 2.5 h immediately before Haiyan's first landfall), the 850-200 hPa vertical wind shear was north-northeasterly at about 4 m $s^{-1}$. Therefore, the downshear-left side is approximately the left-rear side of the track. They found a slight southward vertical tilt (about 1 km from 1- to 7-km altitude) of the center of Haiyan, almost the same as the direction of the vertical tilt. We also

found similar vertical wind shear of about 5 m $s^{-1}$ at a direction of northeasterly, as well a small vertical tilt (about 1 grid box) of the storm center, during the 12-hour period starting at 1200UTC 7 November 2013 (figures not shown). Accordingly, our simulations are in reasonable agreement with observations of Haiyan studied by Shimada et al. (2018).

The next step is to examine the azimuthal averages of the radius distribution for surface fields as well as the vertical structure of the simulated typhoon. For the radius distributions shown later in Figs. 7, 8, 10, and 11, the track lines over a 3-hour



window (the two segments centered at that particular hour, as shown in Fig. 5) were used to identify the semicircles on both sides. Because small deflection of the track tendency are generally found in the 3-hour window, the areas of the right (northern) and left (southern) semicircles are not necessarily the same.

Figure 7 provides Hovmöller diagrams of the 10-m winds, total rainfall rate, and surface enthalpy (latent and sensible heat)
fluxes for 3kmF0. Most noteworthy is the asymmetric structure found in the wind and rainfall fields. The stronger winds over the right (northern) side occurred in association with a larger rainfall rate over the left (southern) side of the typhoon center throughout the integration period. The asymmetric structure can also be found in the surface latent heat flux, whereas it is less evident in the surface sensible heat flux. Similar asymmetric patterns are found in all other simulations (figures not shown). Figure 8 illustrates the composite of radial distributions of selected surface fields for the mature stage, namely
between 1800 UTC 7 November 2013 and 0000 UTC 8 November 2013, for the sensitivity experiments of F0 and F1. The patterns of F2 experiments are generally somewhat in between those of F0 and F1, with their wind fields (enthalpy fluxes) patterns following those of F1 (F0) more closely. This is expected because F1 and F2 use identical momentum transfer coefficient, and F0 and F2 possess similar behavior of the enthalpy transfer coefficients in term of their magnitudes and wind dependent variations. We thereby omitted this subset of experiments in this plot as well as in the rest of this article, for
brevity.

We begin with the composite of 3-km resolution (Fig. 8b). During the mature stage, the radial patterns of mean sea level pressure (SLP) and 10-m winds reveal the typical structure of a tropical cyclone. The difference between the radii of maximum 10-m winds (RMWs) of 3kmF0 and 3kmF1 is insignificant. The tangential wind components dominate the 10-m wind fields. In both experiments, the radii of the maximum radial wind component are slightly larger than those of their
respective tangential winds. In the context of azimuthal averages, the radial distributions of surface enthalpy fluxes are fundamentally constrained by the 10-m wind speed intensity. The aforementioned asymmetric pattern is again evident in the rainfall distribution and even in the wind fields. The 3kmF1 experiment yields a deeper storm central pressure and stronger wind speeds and enthalpy fluxes that the 3kmF0 experiment does. The majority of these increments occur around the eyewall (in terms of the maximum winds). Increasing the grid spacing to 6 km yields a composite radial structure highly
similar to the 3-km group, indicating the relationship among these near-surface fields (Fig. 8c). The major difference between the groups (6 and 3 km) is the size of the inner core with respect to eyewall radius (measured in RMWs) and also the sharpness of peaks in the near-surface fields, consisting with the aforementioned horizontal distributions of these surface fields shown in Fig. 6. As grid spacing decreases, RMWs decrease and the peak sharpness in winds and enthalpy fluxes increases. A much more marked decrease in RMWs and increase in peak sharpness are found in the 1-km resolution group as
compared with the other groups (Fig. 8a). The SLPs, wind fields, and enthalpy fluxes in the 1-km group have narrow but sharp peaks with larger peak values compared with the other resolution groups. The asymmetric pattern of the wind fields is less evident but still can be seen. The 1kmF1 experiment yields a much stronger storm than the 1kmF0 experiment. The asymmetry of the rainfall pattern, however, decreases by using F1. At this point, whether the reduction of grid spacing or the use of more realistic flux formulas leads to a more symmetric (and thus more intense) tropical cyclone remains unclear.





In brief, both the increased resolution and more reasonable flux formulas enhance storm strength. Storm size (in terms of RMWs) is apparently sensitive to the changes in grid spacing but less sensitive to the choice of flux options. A reduction of storm size with smaller grid spacing has been noted by numerous studies (e.g., Davis et al., 2008; Fierro et al., 2009; Gentry and Lackmann, 2010; Kanada and Wada, 2016). As the strength of a storm increases, the eyewall contracts, indicating a

higher efficiency of storm intensification with smaller grid spacing. In our work, the effect of different flux formulas on Typhoon Haiyan simulation is revealed by the changes in storm central pressure and the intensity of maximum winds and enthalpy fluxes around the eyewall. This implies that the underlying mechanisms (i.e., surface flux exchange processes) of various flux options account for storm intensity, whereas the reduction of grid spacing leads to more intense storm structures providing more conducive conditions for enhancing a positive flux effect.

Figure 9 shows the statistics for data points collected within a circle of radius 100 km during a sampling period of 7 hours. From the percentages of binned wind speeds, the major pattern of the differences in flux options is revealed by a rightward shift of the curves in the order of F0, F2, and F1. For each resolution group, a reduction of the higher percentages of moderate to high wind speeds and the emergence of higher wind speeds are observed as comparing the F2/F1 with F0. This feature is more evident in the distributions of 1-km and 3-km resolution groups, in particular for the pattern shift of F1. This

is to say, experiments with F1 can produce extremely high wind speeds (exceeds ~55 m $s^{-1}$) as the resolution increase to 3-km as well as 1-km. For the 6-km resolution, less grid points (< 5%) with wind speed of ~55 m $s^{-1}$ are found even with flux option F1. This can also explain the comparable maximum wind speeds found in the F1 experiments with 3-km and 6-km resolution (cf. Fig. 3c). Therefore the comparable storm intensities, in term of minimum central pressure, found in 3-km and 6-km resolutions (cf. Fig. 3b) are indeed in association with different near surface wind speed distribution (also see Fig. 6).

In 3-km as well as 1-km resolution, more extended and organized areas of extremely high wind speeds are found. Whereas for 6-km resolution, broader areas of high wind speeds with small amount of extremely high wind speed patches are found. Altogether, the wind speed statistics presented here suggest that more detail structures over the full wind speed spectrum can only be resolved as model resolution increase to up to 3-km or higher. This consists with the findings of Davis et al. (2008) and Gentry and Lackmann (2010) that a resolution of approximate 3-km or higher is needed for significant improvement of

intensity simulation of intense hurricanes (the Hurricane Katrina, in their case). With lower resolution, 6-km in our case, we speculate that the momentum energy is diluted over the nearby grid points such that limited magnitude (higher, but not extreme) of wind speeds would become evenly distributed over a broader area. This idea is specifically inspired by the studies of Bryan et al. (2003) and Miyamoto et al. (2013). We have further discussion of the energy distribution of these simulations in next section.

The behaviors of $C_K/C_D$ show the patterns almost strictly follow their respective curves as predicted by the calculation of neutral condition using the pseudo wind data (cf. Fig. 2d). The spread of data, as represented by the range, are rather small and are approximately 10-12% of the magnitude of their respective binned averages. However, the resulting latent heat and sensible heat fluxes present much larger spread of data upon the wind speed bins. For lower wind speeds (and thus smaller mean values of fluxes), the range of data is about 90 % (150%) of the magnitude of the binned mean. For high to extremely



high wind speeds (and thus large mean values of fluxes) the range of data is becoming near or less that 50 % (90 %) of the magnitude of binned mean. We further examined the binned samples and confirmed that the spread in binned data is arising from the discrepancy in stability conditions near the ocean surface. This is expected as referring to eqns. (A2) and (A3). For moderate to high wind speeds, given a value of latent heat or sensible heat flux, the corresponding wind speeds range are

considerable large. Therefore, the structure of near surface moist-air enthalpy is also important for typhoon development. Furthermore, F1 can produce the largest latent and sensible heat fluxes between the three flux options, and the magnitude of the inter-option differences increases as resolution increases. This is obviously because of the larger percentage of extremely high wind speeds that F1 can achieve. In short, the extremely high wind speeds comprise a small percentage of the eyewall circulation of a mature cyclone but have a crucial role in enhancing surface fluxes that supply enthalpy to the eyewall.

As resolution increases, the vertical distribution of tangential winds reveals a significant increase in the vertical extent of stronger wind speed, a reduction of the outward slope of the maximum wind axis, and thus a contraction of the eyewall (Fig. 10). With the change from F0 to F1, tangential wind speeds increase, with an outward and upward expansion of the region of stronger wind speed. No apparent changes are noted in the vertical slope and radius of the eyewall owing to the change in flux options. Both the composite inflow and outflow weaken as resolution increases but enhance as the flux option changes

from F0 to F1. Here the maximum value of inflow (outflow) is 28.3 (−38.8) m s$^{-1}$ in 1kmF0 but is 33.6 (−44.5) m s$^{-1}$ in 1kmF1. Moreover, the maximum cores of tangential winds descend and the depth of inflow decreases as resolution increases. In addition, we can see that the upper level radial inflows are evident in the 1-km resolution group. The upper level radial inflows are apparently weaker in the 3-km resolution group and are hardly found in the 6-km resolution group. The upper level radial inflow above the upper outflow layer is related to the development of upper level warming in the eye; this will be

addressed later in Fig 12.

The contraction of eyewall with reduction of grid spacing is also revealed by a more upright eyewall updraft (Fig. 11). We start with the major eyewall updrafts and downdrafts as denoted by red and blue contours, respectively, in Fig. 11. The reduction of grid spacing also results in stronger updrafts and downdrafts, where the enhanced eyewall downdrafts may be a cause of the decrease in the depth of inflow. The convections outside of the eyewall (weaker upward velocity denoted by

pink hatched area) are apparent in the 1-km resolution group, for which this outside convection in F1 is stronger than that in F0. The outside convection is a result of the azimuthally averaged of the outer rainband. In 3-km resolution group, the outside convections are weaker, but still apparently shown. No outside convection can be clearly distinguished from the eyewall in the 6-km resolution group. This is to say, as resolution decreases, the model may not be able to resolve intense updraft core in the eyewall as well as the outer rainband nearby the eyewall, therefore resulting in a broader area of weaker

updraft. We shall further discuss this in the next section. Stronger inflow and outflow as well as broader eyewall updraft are found at the 6-km resolution, suggesting that secondary circulation is stronger at lower resolution. Fierro et al. (2009) also found a stronger secondary circulation at lower resolution. We further examined the horizontal flow patterns nearby the altitudes of the upper outflow layer, both on model native grids and cylindrical grids, and confirmed that the upper level flow patterns in the 6-km resolution group are more radially outward than the rotational component. As resolution increases, the



upper level flow pattern is becoming more rotational; the radially outward component is suppressed. Because of such discrepancy in the upper level flow pattern, the azimuthally averaged upper outflows (upper level radial winds) in the 6-km resolution group are the strongest between the resolution groups.

The time evolution of the core temperature is represented by the temperature anomaly with respect to a reference mean temperature (Figure 12). Here we term the positive temperature anomaly as warming. The distinct difference found between the resolution groups is the development of the upper level warming layer within the eye. For the 1-km resolution group, there are apparent upper level warming layer appears above ~10 km within a deep warming column in the eye, starting at about 0600 UTC and 0000 UTC on 7 November, respectively for experiments with F0 and F1 (Fig. 12 a,d). The timings of the upper layer warming are associated with their respective upper level radial inflows. The upper level warming layer has long been recognized by observational studies (e.g., Hawkins and Imbembo, 1976) and numerical simulations (e.g., Zhang and Chen, 2012, Chen and Zhang, 2013, Wang and Wang, 2014). In recent years, the formation of the upper level warming layer has been related to the rapid intensification of tropical cyclone (e.g., Zhang and Chen, 2012, Chen and Zhang, 2013, Wang and Wang, 2014). This upper level warming layer, under hydrostatic balance and consider the enhanced effect due to the upper level dryer and thinner conditions of air, can produce much greater impact on the surface pressure falls than the lower-level warming (e.g., Zhang and Chen, 2012). Regarding the formation as well as the maintenance of the upper level warming layer, Zhang and Chen (2012) emphasized the contribution from the high potential temperature air detrained from the lower stratosphere, whereas Chen and Zhang (2013) introduced the contribution from the air detrained from the convective bursts (anomalous intense updrafts) in the eyewall. Both studies revealed the importance of the upper level radial inflow on the warm air detrainment onto the eye region. Wang and Wang (2014) reported that in an environment with decreasing vertical wind shear, air detrained from the lower stratosphere as well as from the convective bursts could result from the development of convective bursts within the inner core region.

From experiment F0 to F1, the upper level warming layer enhances significantly (Fig. 12 a,d). Although both F0 and F1 with 1-km resolution can produce very intense updraft cores, the experiment F1 provides larger energy source (via surface enthalpy fluxes) to the simulated typhoon that experiment F0 does. Therefore, the significantly enhanced upper level warming layer found in experiment F1 could result from the warm detrainment from the intense updrafts (may be seen as convective burst) in the eyewall. A similar development of upper level warming layer can also be found in the 3-km resolution group, but apparently weaker than that in the 1-km group. However, the development of upper level radial inflow appears to precede the main warming for nearly 24-36 hours. This may be partly attributed to the weaker contraction of eyewall in 3-km resolution group. The weaker upper level radial inflow is perhaps another reason for this inconsistence in the time evolution. The 6-km resolution group does not present comparable development of this upper level warming layer as in the 3-km group. This result again indicates that a resolution of 6-km is inadequate to resolve the details of structure and development of a strong typhoon as Haiyan (2013).



## 5 Statistical and energy spectral analyses

Several studies have reported that the reduction of grid spacing yielded deeper, stronger, and more upright and contracted eyewall (e.g., Fierro et al., 2009; Gentry and Lackmann, 2010; Kanada and Wada, 2016). We have also demonstrated that

the size of updraft cores in the eyewall shrank and the region of downdraft increased as resolution is increased. Furthermore, both updraft and downdraft become more intense with the reduction of grid spacing. These features have also been recognized in many studies (e.g., Fierro et al., 2009; Gentry and Lackmann, 2010; Gopalakrishnan et al., 2011; Kanada et al., 2012). In addition, some studies have indicated that large vertical transport of heat and moisture can result in stronger storms given that the model resolution is sufficiently high (e.g., Gopalakrishnan et al., 2011; Kanada et al., 2012).

We utilized contoured frequency by altitude diagrams (CFADs; Yuter and Houze, 1995) to examine the variability of convective cores over the region shown in Fig. 3 (the blue box). CFADs of simulated reflectivity were conducted separately for updrafts and downdrafts (Figs. 13 and 14). In general, the CFADs of reflectivity derived from both updrafts and downdrafts present similar patterns. In these plots, a region bounded by a specific value, say 20%, signifies that 80% of the sampled data points are within this region. Taking the region bounded by the value of 20% as a reference, these plots reveal

that approximately 80% of the sampled data points in all the simulations present an increase in reflectivity with height above approximately 4 km; at heights lower than 4 km, reflectivity demonstrates little change. Similar patterns can be found in studies of CFADs of reflectivity for strong TCs in numerical simulations (Fierro et al., 2009) and for Florida cumulonimbi revealed by observational radar data (Yuter and Houze, 1995). Most noteworthy about our CFADs of reflectivity for both updrafts and downdrafts is the shrinking region bounded by contour of 20% as resolution increases. This indicates that at

higher resolution, the range of reflectivity magnitudes presents a relatively broader distribution, which implies a higher occurrence of higher reflectivity. A descending of the core region of higher frequencies (as bounded by contour of 20%) is found in the CFADs of downdraft compared with those of the updraft. Following the findings of Yuter and Houze (1995), this implies that the larger reflectivity at lower levels may be associated with downdraft and thus can be interpreted as heavy precipitation, whereas higher reflectivity around 4–5 km is mostly associated with updraft and thus may be seen as a

convective cell. However, interpretations are different for reflectivity associated with updraft and downdraft here.

We further conducted CFADs of vertical motions to address this issue (Fig. 15). Most vertical velocities, carried by approximately 95% of the data points as indicated by the region within the 1% contour, are apparently less than 5 m s$^{-1}$ in all simulations. Similar pattern of CFADs of vertical velocity, revealing that most common values of vertical velocity are substantially low, can be found in numerical simulations for TCs (Fierro et al., 2009; Gentry and Lackmann, 2010; Wang

and Wang 2014) and observational data for cumulonimbi (Yuter and Houze, 1995). Furthermore, the region of upward motion is larger than that of downward motion in general. A similar asymmetry of the distributions of upward and downward motion was also evident in simulations by previous studies (Fierro et al., 2009; Gentry and Lackmann, 2010; Wang and





Wang, 2014). In Fig. 15 the most significant differences in the CFADs of vertical velocity are revealed by the lower frequencies, namely the values of 1% and 0.1%. As resolution increases here, both upward and downward motion present an expansion of the distribution of the frequency of 1%. However, the extents of the changes in upward and downward motion with resolution are different: in all simulations, the changes in downward motion with increased resolution are more

significant than those of upward motion. Therefore, the region (occurrence) of downward motion increases as resolution increases. The results also indicate the presence of less downdrafts in lower resolution simulations (i.e., 3 and 6 km), and the degradation of available data points can result in large biases in the CFADs of reflectivity in association with downdraft.

Finally, the 1-km simulations for both F0 and F1 produce the strongest updrafts and downdrafts compared with the 3- and 6-km simulations. Therefore, the higher reflectivity (indicated by the 0.1% contour) in 1-km simulations may be associated

with much more intense updrafts compared with the 6-km simulations. This is consistent with the discrepancy between the secondary circulations of different resolutions in the previous section, where 6-km simulations presented wider and weaker updrafts among the different resolutions. As previously mentioned, the CFADs of reflectivity in the 6-km simulations presented a relatively concentrated range of reflectivity magnitudes compared with the 1-km simulations, implying that the convective cells associated with the eyewalls may be broader in spatial scale. This is confirmed by an examination of

horizontal distributions of the reflectivity of the 6-km resolution of simulations (figures not shown). This can be interpreted as follows. As the resolution is insufficient to resolve the typical cumulonimbus size (~10 km), a number of convective cells (e.g., cumulonimbus clouds) would be interpreted as a broader and perhaps stronger convective plume as discussed in Bryan et al. (2003) and Miyamoto et al. (2013). Bryan et al. (2003) attributed this broader and stronger convective plume to the dilution due to the subgrid-scale mixing. In short, the intense reflectivity (convective) cores of the 1-km simulations are

driven by extreme updrafts occurring within a rather small fraction of the CFAD's computational domain whereas the 6-km simulations are driven by relatively weaker updrafts spread over a much broader spatial distribution.

To a large degree, this resolution dependence of the spatial scale of updrafts may be attributed to the model effective resolution, namely the minimum scale resolved by the discretized model, as discussed by Skamarock (2004). By locating where the simulated energy slope drops below the expected slope, Skamarock (2004) found that the effective resolution of

the WRF model at grid spacing of 4–22 km is generally nearly 7-fold of the model grid spacing (DX). Skamarock (2004) thus indicated that only information at scales greater than around 7 DX in energy spectra represent a physical solution for the WRF model. To examine the effective model resolution in our experiments, here we adopted the algorithm of spectral computation introduced by Denis et al. (2002) for calculation of kinetic energy (KE) spectra. Denis et al. (2002) introduced a two-dimensional discrete cosine transform to convert limited domain of gridpoint atmospheric fields into spectral space. In

this study, the spectral decompositions of velocities (u, v, and w) were first computed for each hourly model output and for each height level, and the KE spectra were then calculated from these velocity spectra. Finally, the KE spectra were averaged over selected periods and layers for Fig. 16. In all simulations, for the KE spectra derived from the horizontal velocity components, we find a model effective resolution of about 7 DX where the values of KE spectra drop significantly, as indicated by Skamarock (2004). Although the vertical velocity spectra exhibit a flatter slope compared with the horizontal





KE spectra, they also present apparent energy drops at wavelengths of approximately 6–7 DX. Our vertical velocity spectra show a pattern (i.e., the slopes) somewhat different from that of Bryan et al. (2003) with respect to the absence of peak values around the respective model effective resolutions. Bryan et al. (2003) conducted simulations with horizontal resolutions much higher than 1 km, namely 500, 250, and 125 m. With their very high resolutions, more intense convections could be resolved. However, Bryan et al. (2003) have indicated that a model resolution of 1 km remains insufficient for explicitly resolving convective clouds. Here, we speculate that insufficient intense convection cores in our simulations may result in the absence of peak values around the respective model effective resolutions.

Finally, the slope of the physical portion of the spectra (for the horizontal and vertical components respectively) remains essentially unchanged as the model resolution is varied. In general, the higher the resolution, the further is the downscale extent, the smaller are the scales represented, and the smaller are model effective resolutions reached. The aforementioned descriptions remain true for all flux options and averaging layers in our work. The key point is that the effective resolution is determined by grid spacing, not the flux options. We noted some subtle differences between the F0 and F1 experiments at smaller wavelengths at nearly 2 DX in the KE spectra, but in only a few hours of the entire simulation period. Therefore, we suggest that although the use of more reasonable flux formulas can increase simulated storm intensity to some extent, the positive effect of the flux formulas cannot be efficiently enhanced unless the grid spacing is appropriately reduced to yield intense and contracted eyewall structure.

## 6 Conclusions

This study investigated the impacts of resolution and surface flux formulas on typhoon intensity and structure simulations through the case study of the Super Typhoon Haiyan (2013). In this study, the effect could be separated between horizontal resolution and air-sea fluxes parameterization through the inner-core structure and spectral analyses. Specifically, we found significant increased sensitivity of TC intensity simulations to surface fluxes parameterizations when model resolution approaches the convective scale (~ 1-km).

Three different sets of surface flux formulas in the WRF model were tested using grid spacing of 1, 3, and 6 km. Both increased resolution and more reasonable flux parameterization could improve typhoon intensity simulation to a certain extent, but their impacts on storm structures were different. A combination of decreasing momentum transfer coefficient and increasing enthalpy transfer coefficients tends to yield stronger storm. The storm size (in terms of the radius of maximum winds) is apparently sensitive to the changes in grid spacing. The choice of flux formulas had little impact on storm size. Sufficiently high resolution was more conducive to the positive effect of flux formulas on simulated typhoon intensity.

Reducing the grid spacing to 1 km yielded a deeper, stronger, and more upright and contracted eyewall. Stronger inflow and outflow as well as wider and weaker eyewall updraft were found in the lower resolutions, indicating a stronger secondary circulation. As strength increased, the eyewall contracted, indicating higher efficiency of storm intensification in smaller grid spacing. This contraction of eyewall is associated with an upper level warming later, which signifies an intense development





of the simulated typhoon. The upper level warming process can be attributed to the air detrained from the intense updraft cores in the eyewall, which can only be resolved by higher resolution, such as 1-km or 3-km.

The analysis of the CFADs revealed that the size of updraft cores in the eyewall shrinks and the region of downdraft increases, and both updraft and downdraft become more intense as resolution is increased. Therefore, the intense reflectivity
(convective) cores of the higher-resolution simulations are driven by more intense updrafts within a rather small fraction of spatial area whereas lower simulations are driven by relatively weaker updrafts spread over a much broader spatial distribution. This resolution dependence of the spatial scale of updrafts is attributable to the model effective resolution based on analysis of KE spectra. The effective resolution is determined by grid spacing, not the flux options. Although the use of more reasonable flux formulas can increase simulated storm intensity to some extent, the positive effect of surface flux
formulas cannot be efficiently enhanced unless the grid spacing is properly reduced to yield intense and contracted eyewall structure.

Despite both updraft and downdraft cores within the eyewall can be partially resolved at 1-km grid spacing, model convergence does not emerge here (e.g., Bryan et al., 2003; Gentry and Lackmann, 2010; Miyamoto et al., 2013). Conducive effect for grid spacing well below 1 km to the contribution of flux parameterization needed to be further explored. Finally,
the typhoon intensity in the experiment 1kmF1 is apparently overestimated. Green and Zhang (2013) suggested that the overestimation of their simulated TC intensity, as compared with the observed best track, may be partially attributed to the neglect of ocean feedback in the model. This is also true in our case. Other components of the numerical model such as boundary layer mixing and the inclusion of wind wave coupling as well as ocean coupling are related to air-sea flux estimates, which is important as ocean feedback (e.g., Davis et al., 2008; Chen et al., 2007; Chen et al., 2010; Zhao et al.,
2017). This is beyond the scope of this paper, but this should be done as well as additional simulation comparisons to other storms for generalizing our results.

**Appendix A**

(a) Fluxes in atmospheric surface layer

In the surface layer, the vertical fluxes of horizontal momentum $\tau$, sensible heat $H$, and latent heat $LH$ near the surface are generally parameterized using bulk flux formulations:

$$\tau = \rho u_*^2 = \rho C_D (U_a - U_s)^2 = \rho C_D (U)^2 \qquad (A1)$$

$$H = -\rho c_p u_* \theta_* = -\rho c_P C_H U(\theta_a - \theta_s) \qquad (A2)$$

$$LH = -\rho L_v u_* q_* = -\rho L_v C_Q U(q_a - q_s) \qquad (A3)$$

where $\rho$ is the air density; $c_p$ is the specific heat capacity of air at constant pressure; $L_v$ is the latent heat of vaporization; $u_*$ is the friction velocity, a velocity scale for the turbulent flow; $\theta_*$ and $q_*$ are the scaling parameters for potential temperature $\theta$ and specific humidity $q$, respectively; and $C_D$, $C_H$, and $C_D$ are the dimensionless bulk transfer coefficients for momentum,





sensible, and latent heat, respectively. The vertical differences in horizontal velocity, temperature, and specific humidity are enclosed in the parentheses in their respective bulk formulas, where the subscripts $_a$ and $_s$ denote the variable 'at a reference height' and 'on the ground or water surface', respectively. Often 10 meters is assumed as the reference height. The three bulk transfer coefficients should all correspond to the same reference height above the surface. Because the wind speed

just on the ground surface is zero, and the surface current over water may be set to zero for sake of simplicity, the bulk formula for vertical flux of horizontal momentum can be reduced to $\rho C_D (U_a)^2$; thereafter we omit the subscript $U_a$ for brevity.

(b) Surface layer scheme in WRF

In WRF, the scaling parameters and bulk transfer coefficients as well as all surface fluxes (momentum, sensible heat, and
latent heat) are parameterized in a surface layer scheme. There are eight surface layer schemes available in the version 3.8.1 of WRF. Seven out of the eight schemes are constructed based on the Monin-Obukhov similarity theory with somewhat different formulations, including those of the roughness lengths, bulk transfer coefficients, and the non-dimensional stability functions defined for wind and potential temperature profiles. Among these available schemes, the surface layer scheme based on the fifth generation Pennsylvania State University–National Center for Atmospheric Research Mesoscale Model
(MM5) parameterization (Grell et al., 1994) has been widely used for a broad range of atmospheric research. In this study, we used the revised version of MM5 surface layer scheme that has been implemented in WRF model prior to version 3.2 (Jiménez et al., 2012). Considering some details of the scheme can change over time, we have compared the corresponding routines in the 3.8.1 version of WRF to the formulations in Jiménez et al (2012) and found some discrepancies. During our writing of the article, we referred to the WRF routines wherever there is a discrepancy found. Below, we briefly describe the
most relevant features of the revised MM5 scheme for our work. Because our focus is on the TC intensity over the ocean, we only document the bulk transfer coefficients over the water surface.

(c) Flux parameterization used in this study

Based upon the Monin-Obukhov flux-profile relationship, the scaling parameters $u_*$ and $\theta_*$ are given as:

$$u_* = \frac{\kappa U}{ln\left(\frac{z+z_0}{z_0}\right)-\psi_m\left(\frac{z+z_0}{L}\right)+\psi_m\left(\frac{z_0}{L}\right)} \qquad \text{(A4)}$$

$$\theta_* = \frac{\kappa(\theta_a-\theta_s)}{ln\left(\frac{z+z_0}{z_0}\right)-\psi_h\left(\frac{z+z_h}{L}\right)+\psi_h\left(\frac{z_h}{L}\right)} \qquad \text{(A5)}$$

where $\kappa$ is the von Karman constant; $L$ is the Obukhov length; $z$ is a specific height level; $z_0$ and $z_h$ are the roughness lengths for momentum and sensible heat, respectively. All roughness lengths are in meters. The Obukhov length $L$ can be calculated from the relation:

$$L = \frac{u_*^2 \theta_a}{\kappa g \theta_*} \qquad \text{(A6)}$$



where $g$ is the gravitational acceleration. The Monin-Obukhov stability functions for momentum ($\psi_m$) and heat ($\psi_h$) are calculated according to variant stability regimes defined in terms of the bulk Richardson number. The details of the stability functions can be found in Jiménez et al (2012). The scaling parameter, as well as the stability function for moisture is assumed to be the same as that for the sensible heat.

The bulk transfer coefficients for momentum and sensible heat are given as follows:

$$C_D = \frac{\kappa^2}{\left[ln\left(\frac{z+z_0}{z_0}\right)-\psi_m\left(\frac{z+z_0}{L}\right)+\psi_m\left(\frac{z_0}{L}\right)\right]^2} \qquad (A7)$$

$$C_H = \frac{\kappa^2}{\left[ln\left(\frac{z+z_0}{z_0}\right)-\psi_m\left(\frac{z+z_0}{L}\right)+\psi_m\left(\frac{z_0}{L}\right)\right]\left[ln\left(\frac{z+z_h}{z_h}\right)-\psi_h\left(\frac{z+z_h}{L}\right)+\psi_h\left(\frac{z_h}{L}\right)\right]} \qquad (A8)$$

The bulk transfer coefficient for latent heat over water surface is given as:

$$C_Q = \frac{\kappa^2}{\left[ln\left(\frac{z+z_0}{z_0}\right)-\psi_m\left(\frac{z+z_0}{L}\right)+\psi_m\left(\frac{z_0}{L}\right)\right]\left[ln\left(\frac{z+z_q}{z_q}\right)-\psi_h\left(\frac{z+z_q}{L}\right)+\psi_h\left(\frac{z_q}{L}\right)\right]} \qquad (A9)$$

where $z_q$ is the roughness lengths for latent heat. Jiménez et al (2012) expressed the stability parameter as $(z + z_0)/L$ in their formulas of $C_H$ and $C_Q$ [cf. their Eqs. (21) and (22)]. In this version of revised MM5 scheme, $(z + z_h)/L$ and $(z + z_q)/L$ are however used for $C_H$ and $C_Q$, respectively. Furthermore, the first term enclosed in the second bracket on the right side of the $C_Q$ formula is mainly valid for land surface as presented in Jiménez et al (2012; cf. their Eq. (22)). In our Eq. (A9), the corresponding term is expressed as $ln\left(z + z_q/z_q\right)$ for water surface according to the WRF routines.

According to the formulations of $C_D$, $C_H$, and $C_Q$, the behaviors of these bulk transfer coefficients are strongly dependent on the momentum and/or scalar (heat and moisture) roughness lengths [cf. Eqs (A7), (A8), and (A9)]. The contribution of momentum roughness length $z_0$ is perhaps more significant than the rest of two. On the right side of the $C_D$ formula, the three terms inside the bracket are expressed in term of $z_0$. Here we term the whole expression with bracket as a drag-dependent effect. A comparison of these formulas indicates that the expression inside the first bracket on the right side of $C_H$

and $C_Q$ formulas are both identical to the expression of drag-dependent effect. In this context, $C_H$ and $C_Q$ shall vary with wind speed due to the drag-dependent effect even with an invariant scalar roughness length.

**Appendix B**

Table B1 provides brief information of the observational estimates shown in Fig 2. For low wind conditions (wind speed

roughly less than $5 \text{ m } s^{-1}$), Vickers et al. (2013) indicated a decrease of $C_D$ with wind speed, in agreement with that observed in COARE 3.0 (Fairall et al., 2003; their Fig. 5), whereas both Geernaert et al. (1987) and Large and Pond (1981)





suggested a nearly constant (no trend) $C_D$. For moderate wind conditions (approximate range of 5-20 m $s^{-1}$), both Large and Pond (1981) and Vickers et al. (2013) indicated that momentum transfer coefficient $C_D$ monotonically increases with wind speed (cf. Fig. 2 a). An upward trend of $C_D$ has also been reported in Geernaert et al. (1987) and French et al. (2007), although the spreads of their respective data are large. This is because shown in Fig. 2 are their data points rather than binned

averages or fitted curves. Three sets of available estimates (Jarosz et al., 2007; Powell et al., 2003; Richter et al., 2016) revealed that the upward trend in $C_D$ would cease under high wind conditions (approximate range of 25-55 m $s^{-1}$), where a level-off or even downward trend would occur instead. According to these observational estimates, the turning points of the wind speed-dependent $C_D$ variations were varied and found within a range of wind speeds of 30 to 40 m $s^{-1}$. Under the much stronger wind conditions (wind speed roughly exceeds 55 m $s^{-1}$), the behavior of $C_D$ again differ considerably from

the relatively lower range of wind speeds. Both Bell et al. (2012) and Richter et al. (2016) suggested a rebound of $C_D$ values with wind speed up to ~70 m $s^{-1}$. However, there are also larger spreads of the $C_D$ values found.

There are fewer observational estimates of heat and moisture transfer coefficient reported in literatures, as compared with the momentum transfer coefficient. Practically, the transfer coefficient for moist-air enthalpy surface flux ($C_K$) were estimated in a number of literatures (e.g., Zhang et al., 2008; Bell et al., 2012; Richter et al., 2016). A similar expression as the eqns. (A2)

and (A3) can be derived for the moist-air specific enthalpy flux, where the specific enthalpy is given as $e = c_p T + L_v q$ (Emanuel, 1995). From a list of literatures that we have referred to, most studies suggested that the bulk transfer coefficients for scalar fields (i.e., sensible heat and moisture as well as enthalpy) are nearly independent of wind speed, with their respective mean values over a range of 0.7 to $1.5 \times 10^{-3}$ (e.g., Geernaert, 1987; Smith, 1988; Drennan et al., 2007; Zhang et al., 2008; Bell et al., 2012; Richter et al., 2016). However, Fairall et al. (2003) reported a steady increase in $C_Q$ with wind

speed based on COARE dataset. Within the range of uncertainty it is plausible to assume $C_Q = C_K$ (e.g., Richter et al., 2016) or even $C_H = C_Q = C_K$ (e.g., Emanuel, 1995; Zhang et al., 2008).




**Table B1** Summary of the previous studies for the observational estimates used in Figure 2. The first column identifies the studies collected for the data extraction, and the source (equation/figure/table) in the corresponding literature. The second column gives the approximate range of the wind speed over which the bulk parameters were estimated; these are not the binned range in some of the literatures. The third column describes the method of flux estimation and the observational data

5 source (field experiments) used in these studies. Abbreviations: ADCP= Acoustic Doppler current profiler, CBLAST=Coupled Boundary Layer Air-Sea Transfer, GPS= Global Positioning System.

| Author (Source in literature) | Wind speed range (*Parameter used*) | Method of flux estimation (Observational data source) |
|---|---|---|
| Bell et al. (2012) (Figures 14, 19 and 20) | 52-72 m $s^{-1}$ ($C_D$, $C_K$, $u_*$) | Absolute angular momentum and total energy budgets (CBLAST experiment; 2003 campaign) |
| French et al. (2007) (Table1)* | 17-40 m $s^{-1}$ ($C_D$, $C_Q$, $u_*$) | Eddy covariance method (CBLAST experiment; 2003 campaign) |
| Geernaert et al. (1987) (Table2) | 3-25 m $s^{-1}$ ($C_D$, $C_H$) | Eddy covariance method (Measurements from Sonic anemometer on North Sea platform) |
| Jarosz et al. (2007) (Fitted equation given in the supplementary) | 20-48 m $s^{-1}$ ($C_D$) | Oceanic momentum budget (ADCP moorings in the northeastern Gulf of Mexico; Hurricane Ivan (2004)) |
| Large and Pond (1981) (Fitted equation given in the abstract) | 4-25 m $s^{-1}$ ($C_D$) | Eddy covariance method and inertial dissipation method (Nova Scotia floating tower and ocean weather station PAPA) |
| Powell et al. (2003) (Figure 3) | 27-51 m $s^{-1}$ ($C_D$, $u_*$) | Flux profile method (GPS dropsonde; 331 wind profiles in the vicinity of 15 hurricane eyewalls) |
| Richter et al. (2016) (Figure 3) | 20-61 m $s^{-1}$ ($C_D$, $C_K$, $u_*$) | Flux profile method (GPS dropsonde; 2425 wind profiles from 37 tropical cyclones) |
| Vickers et al. (2013) (Table 2) | 2-24 m $s^{-1}$ ($C_D$) | Eddy covariance method (Measurements from 11 aircraft datasets; their Table 1) |

*\* The values of $c_Q$ were obtained from Table 1 of French et al. (2007), whereas the corresponding data analysis were*

10 *reported by Drennan et al. (2007).*





**Author contribution**

MTK designed the experiments, analyzed and prepared the manuscript with contributions from all co-authors. WMC, YFS, SCL, and EY performed the simulations, processed and analyzed the model output. TRW, YLT, and CYL joined the
discussions of the study. All authors contributed to the preparation of this paper.

**Competing interests**

The authors declare that they have no conflict of interest.

**Acknowledgments**

This work was financially supported by the Ministry of Science and Technology, Taiwan, under grants 105-2111-M-001-003, 106-2111-M-001-009.

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





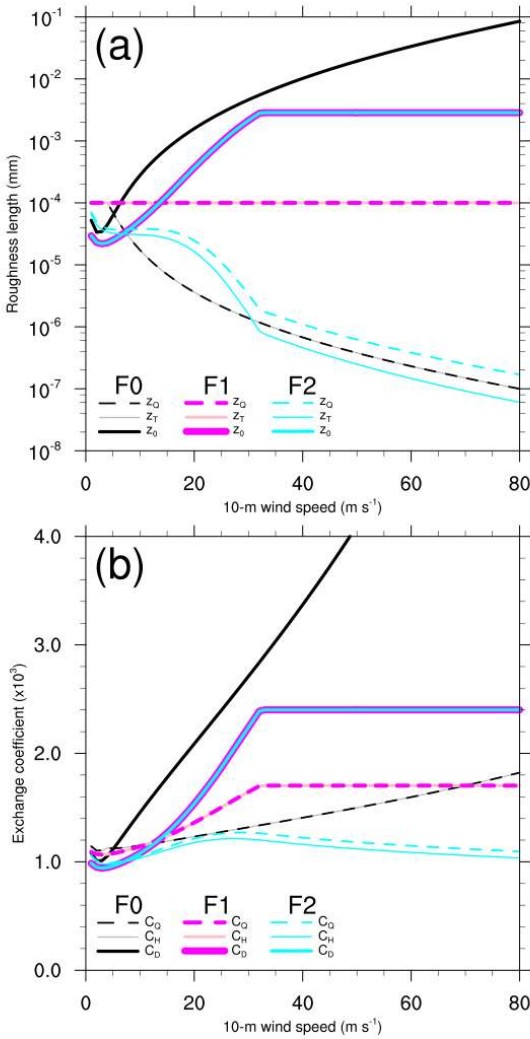

**Figure 1: Plots of (a) roughness lengths and (b) bulk transfer coefficients versus wind speed at 10-m height for neutral condition. Flux schemes F0, F1, and F2 are colored black/gray, red/pink, and green. The thick solid curves are $z_0$ and $C_D$, thin solid curves are $z_h$ and $C_H$, and dashed curves are $z_q$ and $C_Q$. It is noted that some curves are identical. For F1 and F2, $z_0$ (and thus $C_D$) is the same; for F1 and F2 the respective heat and moisture roughness and coefficients are the same: $z_h=z_q$ and $C_H = C_Q$.**





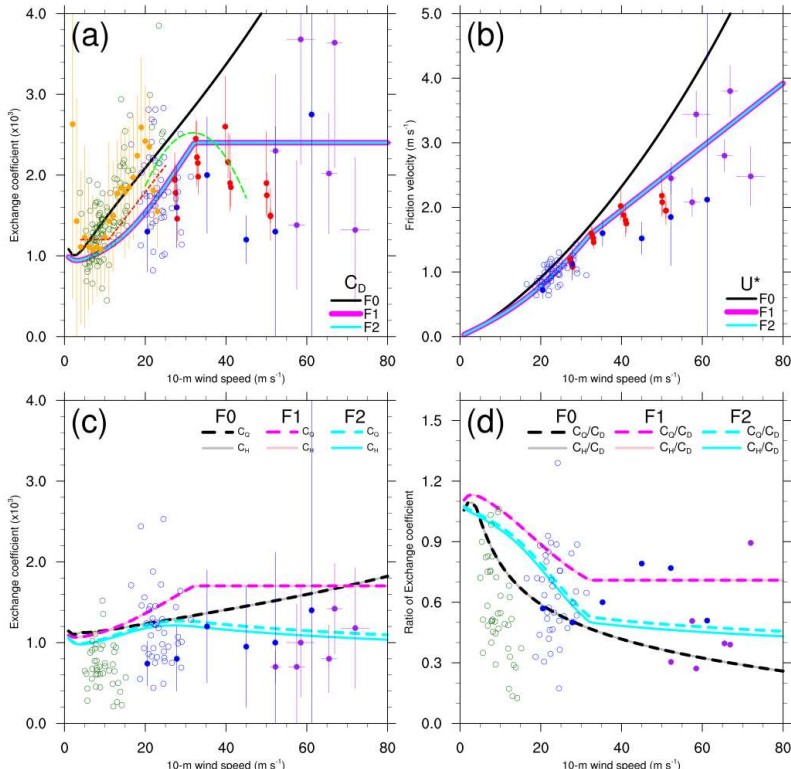

**Figure 2. Plots of (a) momentum transfer coefficients, (b) friction velocity, (c) scalar transfer coefficients, and transfer coefficient ratios versus wind speed at 10-m height from available observational estimates as found in the literatures.**

5 **The scalar transfer coefficients shown here can be $C_H$, $C_Q$ or $C_K$ upon the availability in the literatures that we referred to in this study. The purple dots represent the values from Bell et al (2012); blue circles from French et al. (2007); dark green circles from Geernaert et al. (1987); green curve from Jarosz et al. (2007); red curve from Large and Pond (1981); red dots from Powell et al. (2003); blue dots from Richter et al. (2016), and the orange dots from Vickers et al. (2013). The curves are given by fitted equations; the circles represent the available data points; the dots**

10 **represent average values in a range of wind speed bins with the bars show the corresponding data spread as denoted by either the standard deviations or the 95% confidence intervals, where the wind speed bin and the representation of data spread vary between these literatures. Table B1 provides brief information of these observational estimates.**





The respective bulk parameters derived from WRF, represented as shown in Fig. 1, are superimposed for a comparison.

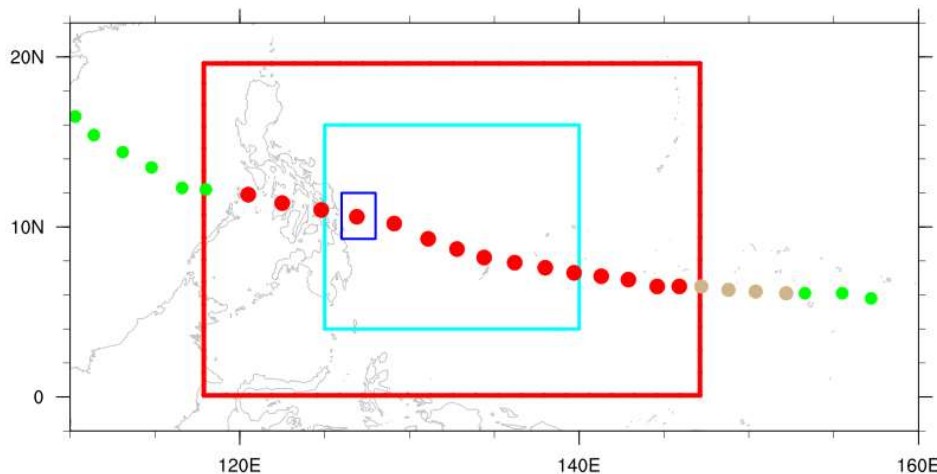

5    **Figure 3. The model domain (red box), and calculation domains for the contoured frequency by altitude diagrams (blue box) and for energy spectra (cyan box). The red, tan, and dots represent the locations of Haiyan obtained from the best-track dataset. The red dots indicate the period of sensitivity experiments, the tan dots indicate the nudging period. Green dots are the locations outside our simulation period.**



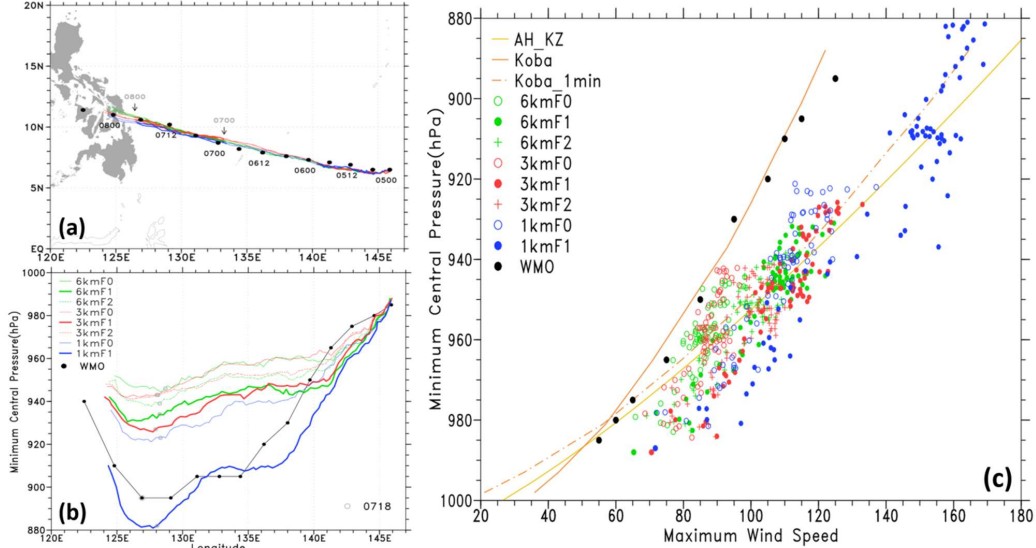

**Figure 4. (a) Haiyan's best track (black dots) and simulated tracks of 1 km (blue lines), 3 km (red lines), and 6 km (green lines) from 0000UTC 5 November 2013 through 1200UTC 8 November 2013. Thick and thin curves represent a span of 24 hours alternately. (b) The evolutions of the best track and simulated minimum central pressures (hPa) shown in relation to longitude. The longitudinal position of the first landfall of Haiyan is around 125°E. The gray circle on each curve denotes the corresponding data point at 1800UTC 7 November 2013. (c) Scatterplot of maximum wind speed versus minimum central pressure. Reference curves Koba and AH_KZ are the operational WPRs for the JMA and JTWC, respectively. Koba_1min is the modified Koba curve with 10-minute sustained winds converted to 1-minute averaged values.**




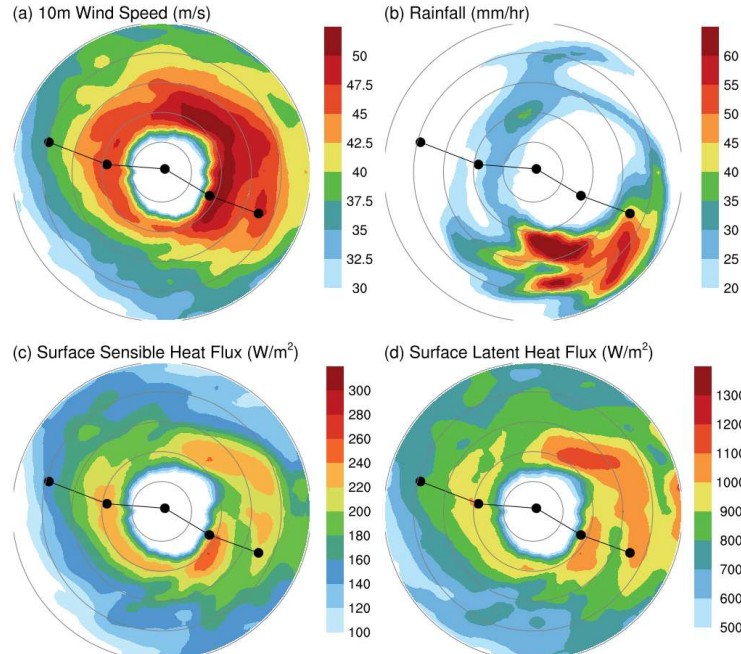

**Figure 5. Horizontal distributions for (a) wind speed ($m\ s^{-1}$) at 10-m height, (b) total rainfall ($mm\ hour^{-1}$), (c) surface sensible heat flux ($W\ m^{-2}$), and (d) surface latent heat flux ($W\ m^{-2}$) at forecast hour 71 h (valid at 2200UTC 7 November) for the experiment F0 with 3-km resolution. All fields are taken from the transformed subset on cylindrical grids; see text for detail. At 71 h, the simulated typhoon centered at the nearest location upon the best-track location at 1800UTC 7 November 2013. Range rings are shown at radii of 20, 40, 60, 80, and 100 km. Black dot at the plane center denotes the simulated typhoon center at 71 h, whereas the two dots to the upper left and lower right of this central dots are the corresponding locations of typhoon center in the next and previous 2 hours. Solid lines connecting these dots denote the simulated typhoon track within the 5 hours.**



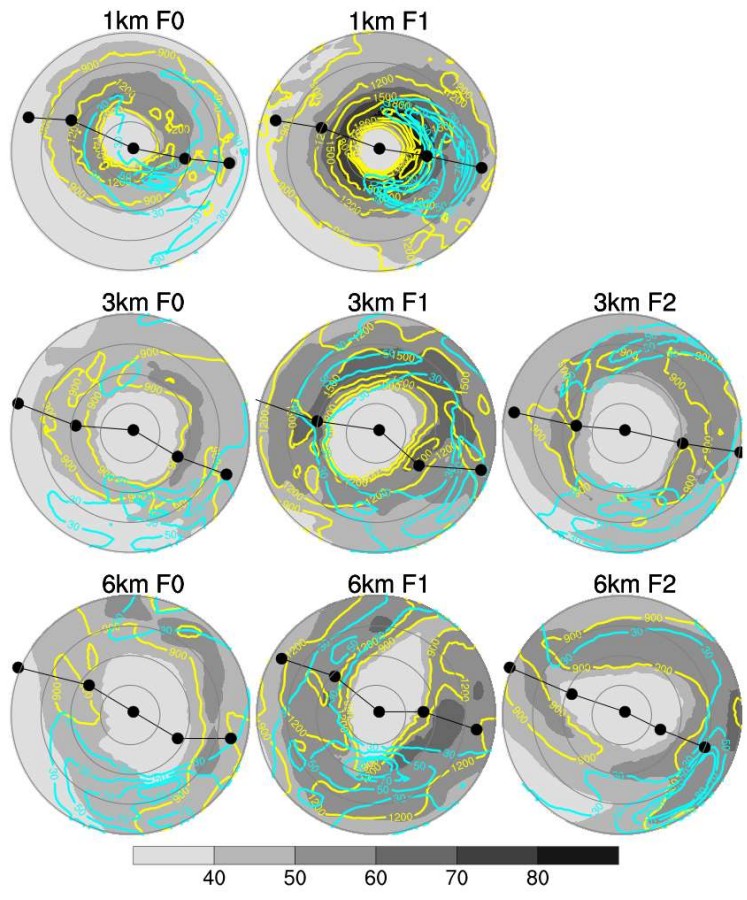

**Figure 6. Horizontal distributions for wind speed at 10-m height (shaded; $\mathrm{m}\ s^{-1}$), surface latent heat flux (yellow contours; $\mathrm{W}\ m^{-2}$), and total rainfall (cyan contours; $\mathrm{mm}\ hour^{-1}$) at 71 h for all experiments. In all the experiments, the center location of simulated typhoon at 71 h are at the nearest or the second nearest location upon the best-track location at 1800UTC 7 November 2013. The distributions for experiment F0 of 3-km resolution are the same as in Fig.5 but slightly zoom in on a circle of radius 80 km; Range rings are shown at radii of 20, 40, 60, and 80 km. Other symbols are shown as in Fig. 5.**




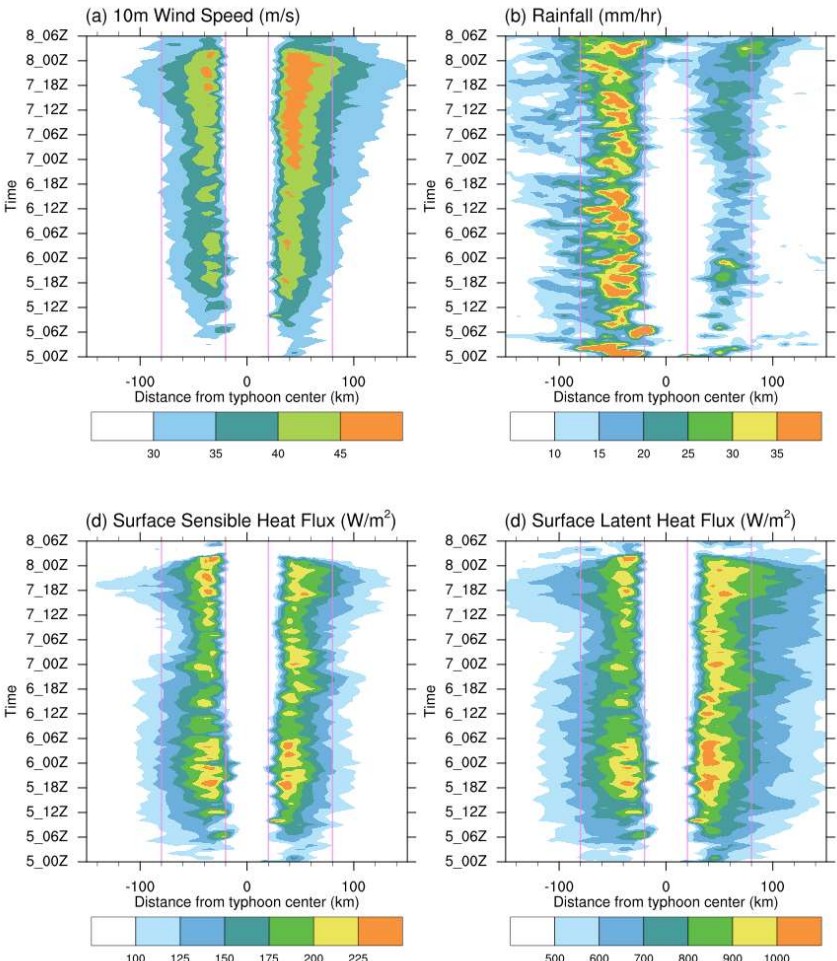

**Figure 7. Radius–time cross sections of (a) 10-m wind speed (m s⁻¹), (b) rainfall (mm hour⁻¹), (c) surface sensible heat flux (W m⁻²), and (d) surface latent heat flux (W m⁻²) for control experiment (3kmF0). The positive and negative distances from typhoon center represent the azimuthal averages of the right (northern) and left (southern) semicircles, respectively, along the typhoon track. For each hourly output, the corresponding track in 3 hours centered at that particular hour is used to identify the semicircles on both sides.**



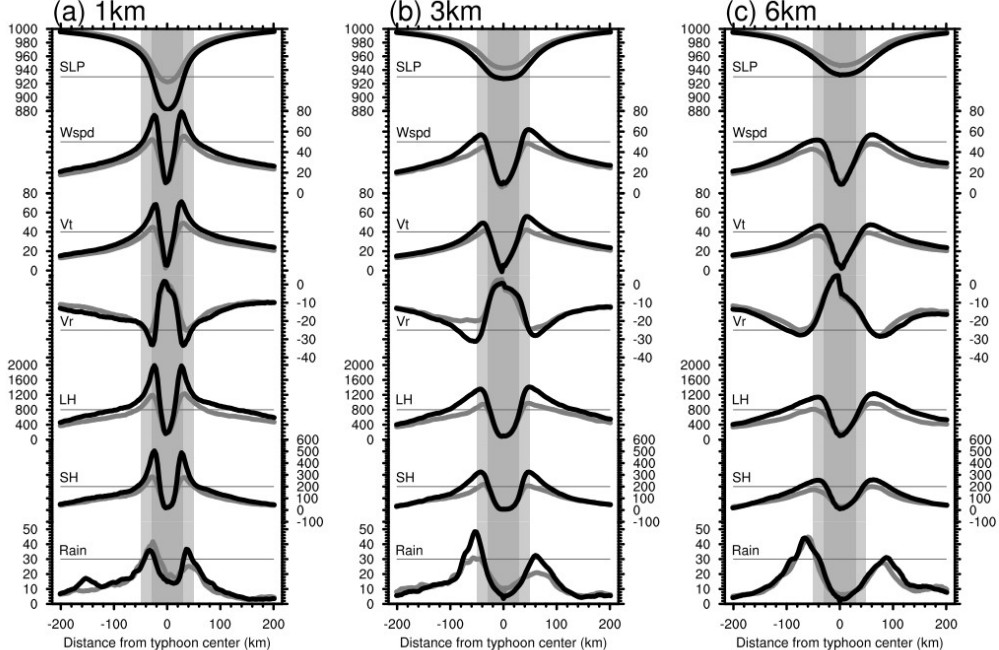

**Figure 8. The radial variations of surface fields for experiments 1 km (left), 3 km (center), and 6 km (right). The surface fields shown are (from top to bottom) sea level pressure (SLP; hPa), wind speed and tangential and radial winds at 10 m (Wspd, Vt, and Vr; m s⁻¹), surface latent and sensible heat fluxes (LH and SH; W m⁻²), and total rainfall (Rain; mm hour⁻¹). Positive values of Vr indicate inflow. Dim-gray and black curves represent flux options F0 and F1, respectively. Horizontal gray lines are selected reference lines for each field. All fields are time averaged between 1800UTC 7 November 2013 and 0000UTC 8 November 2013. Positive and negative distances represent the right (northern) and left (southern) semicircles as in Fig. 7. Vertical light-gray and gray shaded bars denote the radial sectors bounded by radii of 50 km and 30 km, respectively.**




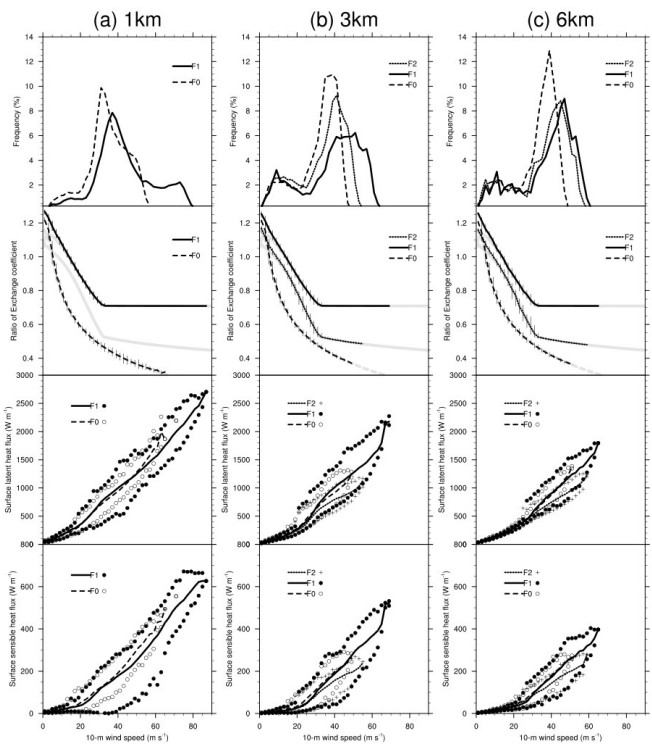

**Figure 9.** The top panels are the percentage of samples in $2\ \mathrm{m}\ s^{-1}$ bins for wind speeds. These wind speed samples were collected inside a circle of radius 100 km with respect to the typhoon center at each output hour during a sampling period from 1800UTC 7 November 2013 to 0000UTC 8 November 2013. Over this sampling period of 7 hours, only points over ocean are collected. The percentages were the numbers of data point in each wind speed bin divided by the total count during the sampling period. The upper middle, lower middle, and bottom panels show the binned averages and data ranges for ratio of $C_K/C_D$ at 10-m height, latent heat flux, and sensible heat flux, respectively, in each $2\ \mathrm{m}\ s^{-1}$ wind speed bins. Here the values of $C_Q$ at 10-m height is used as $C_K$. The binned averages are the mean values for the corresponding variable as collected in bins depending on the collocated wind speed and are displayed as solid or dashed curves. Within each wind speed bin, the respective data range for each variable is bounded by the minimum and maximum values from all binned samples during the period of 7 hours. The data ranges are denoted as vertical bars ($C_K/C_D$) and symbols (fluxes). Gray curves in the panels of $C_K/C_D$ are the same as in Fig. 2d, superimposed for a reference.





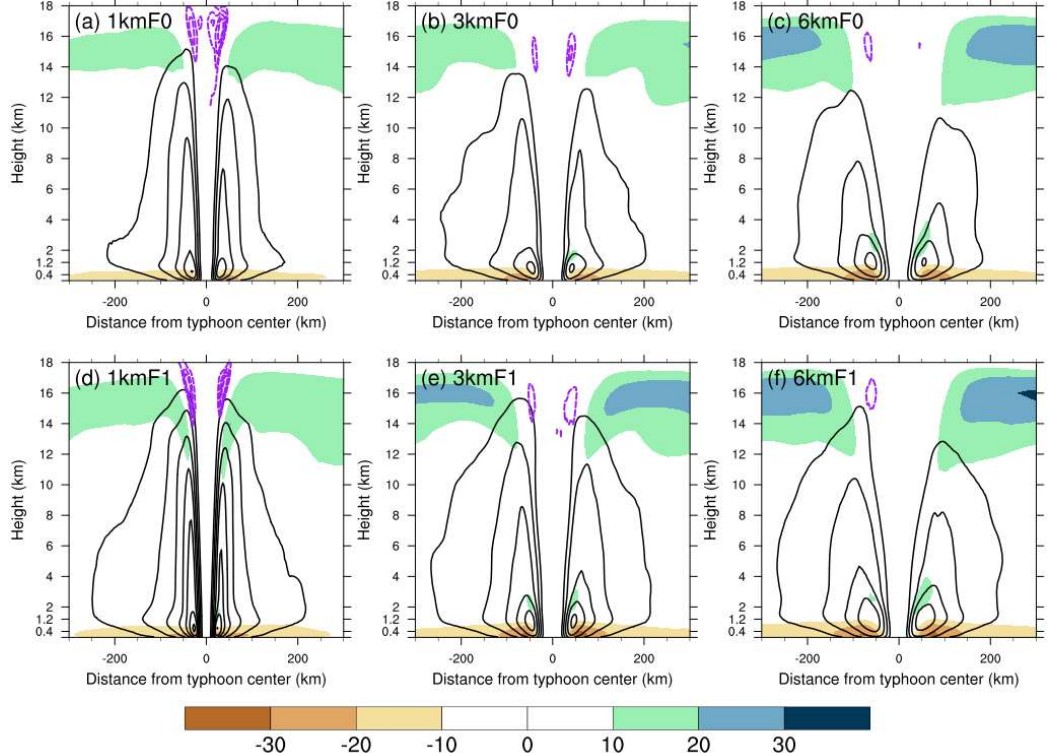

**Figure 10. Radius–height cross sections of tangential wind (contours; m s⁻¹) superposed on radial wind (color shaded; m s⁻¹) for experiments (a) 1kmF0, (b) 3kmF0, (c) 6kmF0, (d) 1kmF1, (e) 3kmF1, and (f) 6kmF1. The tangential winds are contoured every 10 m s⁻¹ starting from 30 m s⁻¹. Positive values of radial wind indicate inflow. The upper level radial inflows (purple dashed contours; $m\ s^{-1}$) are also imposed and are contoured every 0.5 $m\ s^{-1}$ starting from -0.5 $m\ s^{-1}$. All fields are time averaged between 1800UTC 7 November 2013 and 0000UTC 8 November 2013. Positive and negative distances represent the right (northern) and left (southern) semicircles as in Fig. 7.**





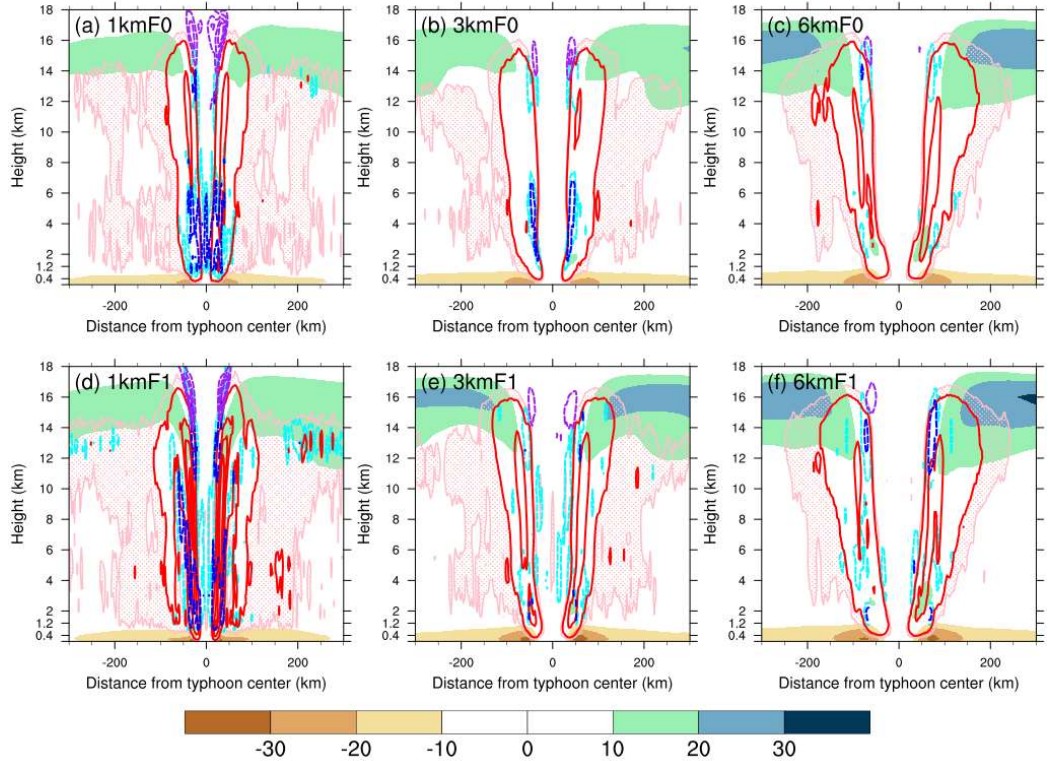

Figure 11. The same as Fig. 10 but for vertical velocity (contours; m s⁻¹) and radial wind (color shaded; m s⁻¹).
Positive (upward) and negative (downward) velocities are indicated by red and blue contours, respectively, and are
plotted every 1 m s⁻¹ starting from wind speeds of 0.5 m s⁻¹. In addition, pink hatched areas denote upward velocity
of 0.3-0.5 $m\ s^{-1}$; cyan contours are downward velocity of -0.3 $m\ s^{-1}$.





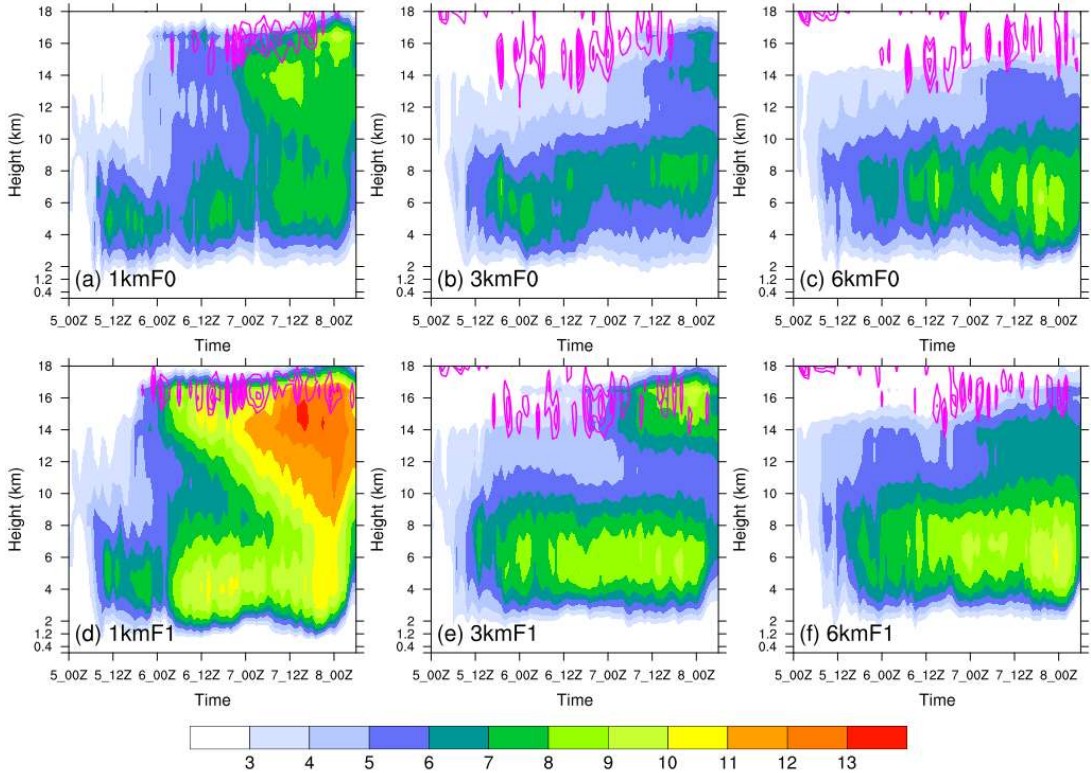

**Figure 12. Time-height cross sections of upper level radial inflow (contours; $m\ s^{-1}$) superposed on temperature anomaly (color shaded; $K$) for experiments (a) 1kmF0, (b) 3kmF0, (c) 6kmF0, (d) 1kmF1, (e) 3kmF1, and (f) 6kmF1. The upper level radial inflows are contoured every 0.5 $m\ s^{-1}$ starting from -0.5 $m\ s^{-1}$. The temperature anomaly is defined with respect to a composite of domain-averaged temperature profiles at the model initial time of the sensitivity experiments (0000UTC 5 November 2013); the composite of domain-averaged temperature profiles is derived as the mean of 8 domain-averaged temperature profiles obtained from all the experiments. The time-height cross section of temperature anomaly is the area-averaged temperature profile inside a ring of radius 21 km with respect to the typhoon center at each output hour minus the composite of domain-averaged temperature profiles. The upper level radial inflows are the area-averaged radial winds inside a ring of radius 200 km with respect to the typhoon center at each output hour; only the radial winds above 9 km height are used for calculation.**

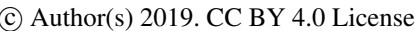



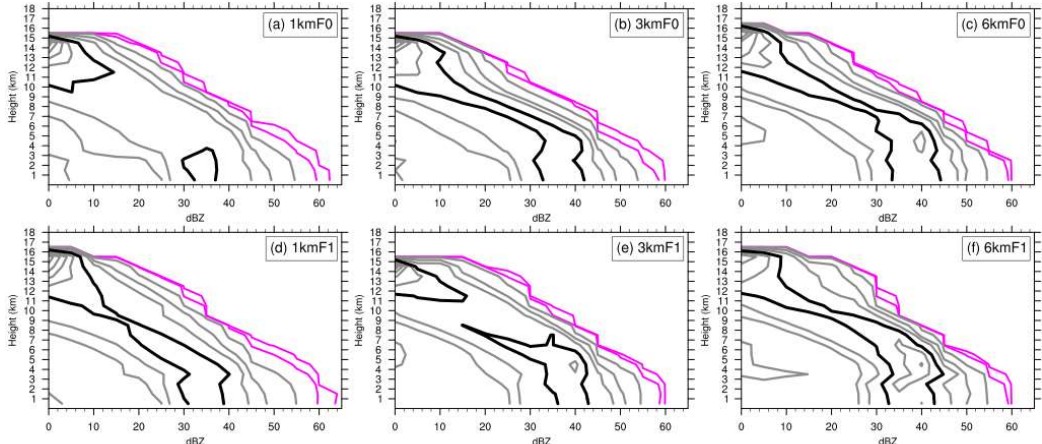

**Figure 13. The CFADs of simulated reflectivity for experiments (a) 1kmF0, (b) 3kmF0, (c) 6kmF0, (d) 1kmF1, (e) 3kmF1, and (f) 6kmF1. The sampling time for model output are taken as the simulated typhoon centered at its nearest location upon the best-track location at 1800UTC 7 November 2013 and for the following 2 hours afterwards. The simulated reflectivity was taken from all grid points with updraft (positive vertical velocity) within the calculation domain for CFADs (see blue box in Fig. 3 for the corresponding area) in the model native grids. For reflectivity CFADs, the bin size is 5 dBZ and the plot is contoured at 0.01%, 0.1%, 1%, 5%, 10%, 20%, 30%, 40%, 50%, and 60% of data per dBZ per kilometer, with the 20% dBz$^{-1}$ km$^{-1}$ contour highlighted in black and the lowest 2 contours displayed in magenta.**



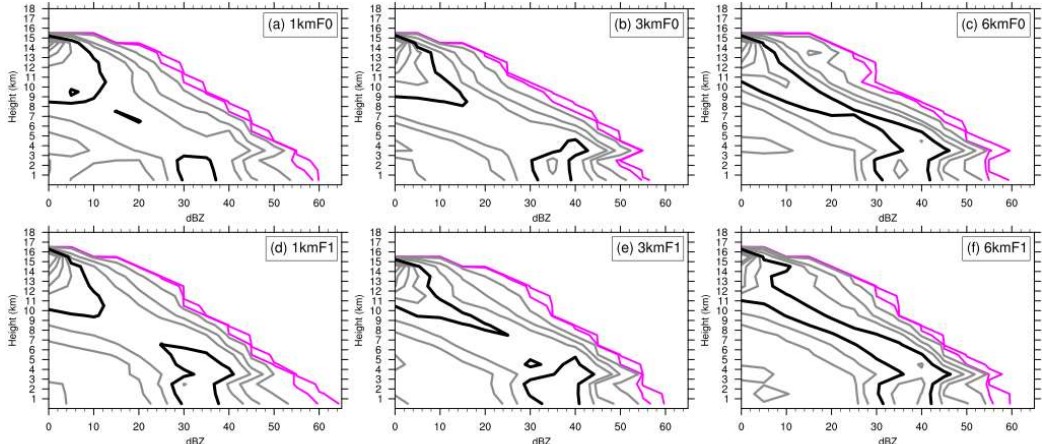

**Figure 14. The same as Fig. 13 but for CFADs of simulated reflectivity with downdraft.**

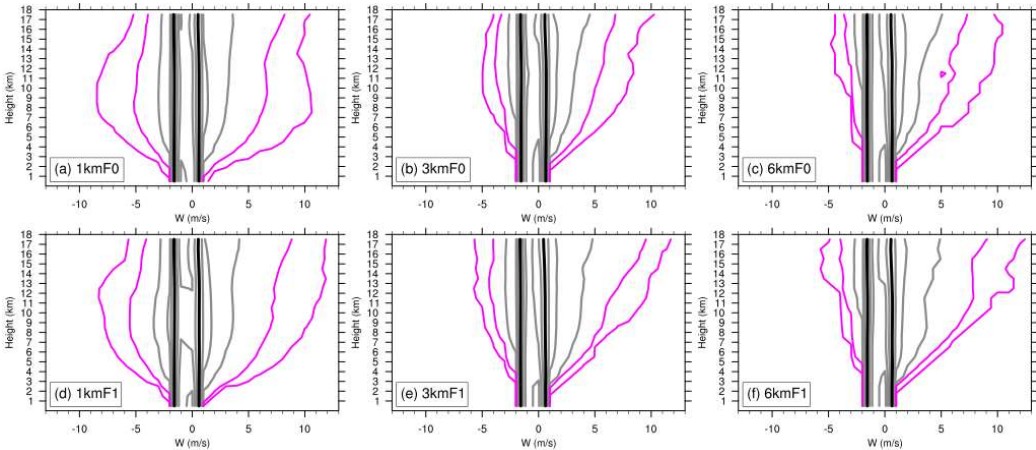

5 **Figure 15. The same as Fig. 14 but for CFADs of simulated vertical velocity. The velocity bin size is 1 m s$^{-1}$ and the plot is contoured at 0.01%, 0.1%, 1%, 5%, 10%, 20%, 30%, 40%, 50%, and 60% of data per meter per second per kilometer, with the 20% (m s$^{-1}$)$^{-1}$ km$^{-1}$ contour highlighted in black and the lowest 2 contours displayed in magenta.**


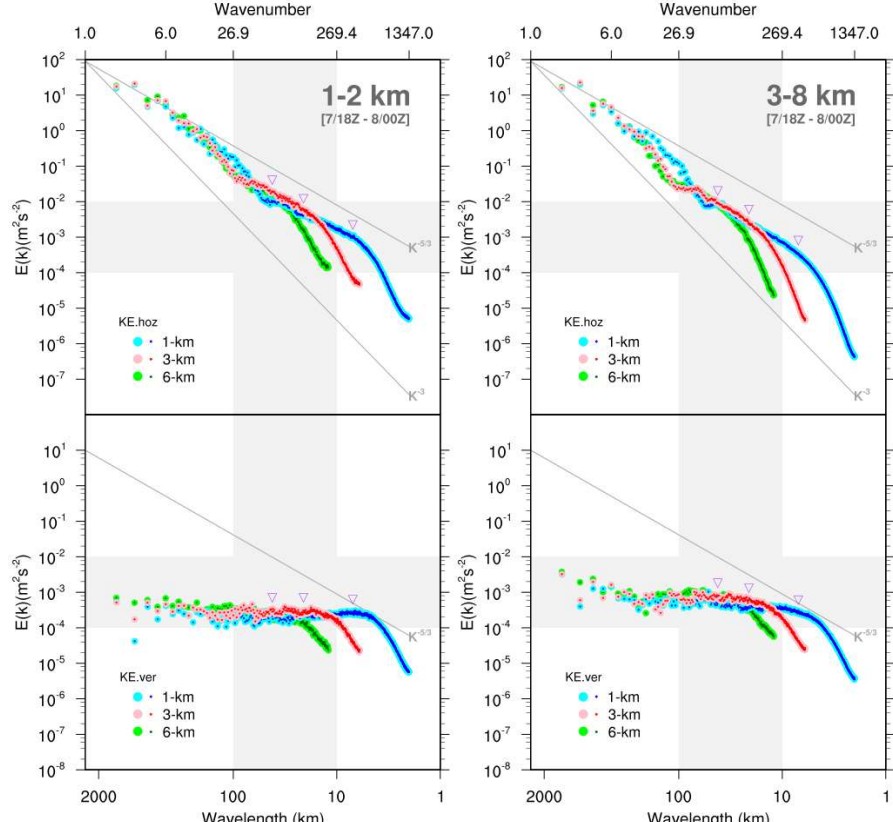

**Figure 16. Mean KE spectra for horizontal wind (upper) and vertical velocity (lower) for all simulations as computed over the boundary layer (left panel; between 1 and 2 km) and free troposphere (right panel; between about 3 and 8 km) for the mature stage. The mature stage is indicated by the average between 1800UTC 7 November 2013 and 0000UTC 8 November 2013. The gray lines correspond to the expected slopes for the large scale (−3 slope) and mesoscale to smaller scales (−5/3 slope) of the atmospheric kinetic energy spectrum. The wavelength and wavenumber are given on the lower and upper x-axes, respectively. The purple triangles in each panel, from left to right, denote the locations of 7 DX for the simulations of resolutions 6 km, 3 km, and 1 km, respectively. The domain of the spectral decomposition is shown in Fig. 3.**



**Table 1 Summary of the resolutions and surface flux options tested. The nudging period for all experiments is between 0000 UTC 4 November 2013 and 0000 UTC 5 November 2013, with identical nudging setting for each resolution group using the default flux option (F0). Thereby the corresponding sensitivity experiments start with identical initial TC (tropical cyclone) structure and location. The MCP and MSW indicate, respectively, the minimum central pressure at TC center and maximum surface wind speed near TC center.**

| Experiment Name | Resolution (km) | Flux Option | Notes |
|---|---|---|---|
| 6kmF0 | 6 | 0 | Nudging stage: model resolution and flux option configured as 6kmF0 |
| 6kmF1 | 6 | 1 | Initial TC: MCP=988.0hPa and MSW=33.7 m s$^{-1}$; 145.79°E, 6.36°N |
| 6kmF2 | 6 | 2 | Horizontal grid points: 538 x 367; Time step: 20 s |
| 3kmF0 | 3 | 0 | Nudging stage: model resolution and flux option configured as 3kmF0 |
| 3kmF1 | 3 | 1 | Initial TC: MCP=987.6hPa and MSW=36.3 m s$^{-1}$; 145.95°E, 6.35°N |
| 3kmF2 | 3 | 2 | Horizontal grid points: 1075 x 733; Time step: 10 s |
| 1kmF0 | 1 | 0 | Nudging stage: model resolution and flux option configured as 1kmF0 |
| 1kmF1 | 1 | 1 | Initial TC: MCP=986.5hPa and MSW=36.9 m s$^{-1}$; 145.88°E, 6.46°N |
| | | | Horizontal grid points: 3223 x 2197; Time step: 3 s |

