# Peer review of "Effects of Horizontal Resolution and Air–Sea Flux Parameterization on the Intensity and Structure of simulated Typhoon Haiyan (2013)"

_Natural Hazards and Earth System Sciences, 2018_

## Referee Comment (RC1) · Anonymous Referee #1 · 19 Feb 2019

GENERAL COMMENTS:

Through numerical simulation experiments and analysis, the authors present a study aimed at evaluating the influence of horizontal resolution and surface flux formulas on the development of Typhoon. The results show the increase of resolution and more reasonable surface flux formulas leads to an improvement of the typhoon intensity simulation. Although the innovation of the paper is not strong, the relevant conclusions of the paper have a positive impact on typhoon forecast.

It is clear a lot of effort has gone into this research. However, I question the design of the sensitivity experiments. As such, the recommendation is for major revisions, and

this reviewer feels this manuscript will be improved by addressing the comments below.

SPECIFIC COMMENTS

1. On P8 line 5-9, "Because the simulation result of F2 is somewhat between those of F0 and F1 for other resolutions, we omitted the F2 test at 1-km resolution." However, judging from the time evolution of typhoon intensity, the observed typhoon intensity is located between 1km F0 and 1km F1 experiments. Is there a better coincidence between 1km F2 and observation?

On P7 line 20-25, the author also mentioned that "F1 predicts larger CH and CQ at all wind speeds than F2 does, implying that F1 has a potentiality to gain larger enthalpy fluxes", This may also be the reason why the simulated typhoon intensity of 1km F1 is higher than that observed.

Overall, I think the numerical test of 1 km F2 is very critical and cannot be omitted.

2. On P4 line 5, the horizontal grid spacing is 1, 3 and 6 km. So why do the authors still use the updated Kain-Fritsch convective scheme? If the convective scheme is not used, do the simulation results change?

3. On P8 line 10, the general simulation usually chooses to gradually increase the resolution through nesting, while all resolution simulations in this paper do not use nesting. Why?

4. Section 2 should be divided into subsections, such as 2.1 moisture roughness length, 2.2 momentum roughness length, 2.3 bulk transfer coefficients, 2.4 experimental designs.

5. On P8 line16-17, please show the original resolution of GFS data used.

6. From Fig. 4a, it is difficult to distinguish the track difference between different sensitivity tests, and it is suggested to modify this figure.

7. On P9 Line 6, "The intensity is not very sensitive to the resolutions of 3 and 6 km...",

while I think it is sensitive enough.

8. There is no observed typhoon structure in this article as a comparison, and it is suggested to add corresponding figures.

9. The authors can introduce the use of contoured frequency by altitude diagrams. Maybe some readers are not familiar to those figures.

10. In general, this quality of this paper is a little difficult to be understood by readers. I hope the authors can carefully revise some long sentences to make it easy to understand.

11. Why doesn't the positive effect of reasonable surface flux formulas be enhanced efficiently if the grid spacing is relatively large? It should explain more clearly in the paper.

TECHNICAL CORRECTIONS

1. Page 3, Line 27, change "reaches" to "reached"

2. Page 7, Line 29, change "were" to "was"

3. Page 7, Line 31, change "level-off" to "level-offs"

4. Page 8, Line 3, change "between" to "among"

5. Page 8, Line 4, this sentence is incomplete.

6. Page 9, Line 30, change "agree" to "agrees"

7. Page 11, Line 18, change "enhanced" to "enhancement"

8. Page 12, Line 29, change "increases" to "increase"

9. Page 13, Line 22, change "detail" to "detailed"

10. Page 13, Line 23, change "increase" to "increases"

11. Page 15, Line 5, change "between" to "among"

12. Page 15, Line 13, change "consider" to "considering"

13. Page 15, Line 24, change "that" to "than"

14. Page 17, Line 26, change "represent" to "represents"

15. Page 18, Line 21, change "fluxes" to "flux"

16. Page 18, Line 32, change "later" to "layer"

17. Page 19, Line 19, change "as" to "for"

---

## Referee Comment (RC2) · Anonymous Referee #2 · 11 Mar 2019

This contribution aims at investigating the impacts of model horizontal resolution and surface flux formulas on structure and intensity of tropical cyclones. The authors intend to study the sensitivity of the intensity of tropical cyclones to surface fluxes parameterizations of the Weather Research and Forecasting Model (WRF) at very high resolutions (1, 3 and 6 km, when model resolution approaches the convective scale).

The paper is in general structured and scientifically sound. English should be revised by a native speaker in order to correct some typos, but also to rephrase long sentences that make the manuscript hard to read. Scientifically, there are several limitations, especially regarding the design of the simulations, that require major revisions of the

submitted version.

1. Is convection parameterized (K-F cumulus scheme) for all resolutions? 6 km simulations are in the grey-zone, but cumulus should be explicitly resolved when working at resolutions below 3-4 km. How this parameterization impacts the results?

2. P. 4: What is the nesting approach followed in the simulations? Is it a one way or a two way nesting?

3. Nudging is critical for correctly representing the TC structures, their paths and intensities. So, ideally, some sensitivity runs should take nudging options into account. If not possible, further details about nudging approach followed and its impacts on the results should be elaborated.

4. P. 8, lines 5-10: It is not clear to the reader why 1 km F2 test is omitted. The authors indicate that the "the simulation result of F2 is somewhat between those of F0 and F1 for other resolutions, we omitted the F2 test at 1-km resolution". Can you extrapolate that for the 1 km resolution? I think the authors should elaborate on F2 at 1 km.

5. I cannot find the observed structure of the TC in the manuscript, which would be very important for building confidence on the modelling results.

---

## Author Comment (AC2) · 16 Apr 2019

Effects of Horizontal Resolution and Air–Sea Flux Parameterization on the Intensity and Structure of simulated Typhoon Haiyan (2013)

Mien-Tze Kueh, Wen-Mei Chen, Yang-Fan Sheng, Simon C. Lin, Tso-Ren Wu, Eric Yen, Yu-Lin Tsai, Chuan-Yao Lin

**Response to referee 2**

Dear Referee,

We thank the referee for the comments and suggestions to our manuscript. Please find our responses to your comments and questions below. The comments and questions are given in italics, and our responses are in blue.

*This contribution aims at investigating the impacts of model horizontal resolution and surface flux formulas on structure and intensity of tropical cyclones. The authors intend to study the sensitivity of the intensity of tropical cyclones to surface fluxes parameterizations of the Weather Research and Forecasting Model (WRF) at very high resolutions (1, 3 and 6 km, when model resolution approaches the convective scale).*
*The paper is in general structured and scientifically sound. English should be revised by a native speaker in order to correct some typos, but also to rephrase long sentences that make the manuscript hard to read. Scientifically, there are several limitations, especially regarding the design of the simulations, that require major revisions of the submitted version.*

We thank the referee's positive comments to our manuscript. We have carried out the numerical experiment of 1 km F2 in response to your comment 4. The revised manuscript has been edited by *Wallace Academic Editing* and is considered to be improved in grammar, punctuation, general readability, and native English usage. In addition, we followed the suggestion of *Wallace Academic Editing* and reworded the "impacts" to "effects" in our title, which is now:

> Effects of Horizontal Resolution and Air–Sea Flux Parameterization on
> the Intensity and Structure of simulated Typhoon Haiyan (2013)

Below are our point-by-point replies to your comments.

1. *Is convection parameterized (K-F cumulus scheme) for all resolutions? 6 km simulations are in the grey-zone, but cumulus should be explicitly resolved when working at resolutions below 3-4 km. How this parameterization impacts the results?*

Yes, the convection parameterization (K-F cumulus scheme) was used for all resolutions. This is because we intend to keep consistency among all cases. The convective treatment was also mentioned in the comment 2 by referee 1. For typhoon Haiyan, we have carried out several tests with and without cumulus parameterization on the 3 km resolution grid. The 3 km resolution grid is for our control experiment in this study. The tests revealed that, simulations with and without cumulus parameterization produced overall similar simulated storm intensity. We did not perform any convection parameterization test on the 1-km resolution grid, because of our limited computational resource.

Numerous studies have suggested that 3-4 km resolutions without any cumulus parameterization is sufficient to represent mesoscale convections (e.g.,Weisman et al. 1997; Davis et al. 2008; Gentry and Lackmann, 2010). However, such a grid resolution is still insufficient for representing individual convective cells (e.g., Bryan et al. 2003; Miyamoto et al. 2013). The use of cumulus parameterization with the grid spacing below 3-4 km has been investigated by a number of recent studies. Some studies suggested to activate the cumulus parameterization for simulation of moist convective event with a grid resolution of 4 km (Deng and Stauffer 2006), 3 km (Lee et al 2011) and 2 km (Kotroni and Lagouvardos 2004). Some others, however, revealed that the activation of cumulus parameterization for simulation with grid spacings of 2-3 km produced overall similar simulated storm as in the simulation with explicit convection (e.g., Yu et al. 2011; Li et al. 2018; On et al. 2018). Sun et al. (2013) studied the appropriateness of a variety of cumulus parameterization schemes used in high-resolution simulations. They assumed that the cumulus scheme is closely related to the model convergence in simulating TC intensity. Here, a convergence of model solution in terms of TC intensity is that the simulated TC intensity would remain similar irrespective of any further reduction of the grid spacing. They found a weak convergence in fine resolution (from 3 to 1 km) simulations with most of the schemes, whereas the convergence is relatively strong in the simulations with a scale-aware scheme designed for any resolution. Accordingly, cumulus parameterization may still play a role in the fine resolution (3 to 1 km) simulations. The question then arises as to what is the appropriate design of cumulus parameterization for very high resolution. However, this is far beyond the scope of our present study.

2. *P. 4: What is the nesting approach followed in the simulations? Is it a one way or a two way nesting?*

No nesting approach was used for our simulations in this study. Our large single domain was chosen to cover the majority of simulated Haiyan (2013)'s convection during the period of sensitivity simulation, and to make a cleaner comparison among those experiments running at different resolutions.

Higher-resolution nested model configuration are widely used in numerical weather prediction and regional climate modelling. The main reason for this is because large area of high-resolution model simulation is computationally too expensive. However, consistency between nested grids is also important. With lateral boundaries on multiple grids, model solutions may not be smooth across nested-domain boundaries. In a nested WRF simulation, a discontinuity in precipitation and moisture fields (i.e., a sharp gradient) across the inner domain boundaries has long been recognized by the WRF community. Uncertainty related to the use of multiple nested grids can resulted from mismatched model physics across nested-domain boundaries. For example, Warner and Hsu (2000) revealed that the treatment of convection on the outer grid can affect the explicit convection on the inner grid. Their result indicated that the simulation biases related to the parameterized convection (e.g., errors in precipitation timing, precipitation intensity, and the vertical distribution of latent heating) can greatly modulate the explicit convection on the inner grid through the induced subsidence from the outer grid. The nesting issue was also mentioned in the comment 3 by referee 1.

3. *Nudging is critical for correctly representing the TC structures, their paths and intensities. So, ideally, some sensitivity runs should take nudging options into account. If not possible, further details about nudging approach followed and its impacts on the results should be elaborated.*

In this study, the simulations were performed with two stages. The nudging stage is a 24-hour period before the 78 hours sensitivity stage. No analysis nudging was applied to the model grid during the entire period of sensitivity experiment.

In this study, the analysis nudging was applied to the horizontal wind components, potential temperature, and water vapor mixing ratio. The nudging coefficients for all variables were set at 0.0003 $s^{-1}$. The nudging was only applied at all levels above the planetary boundary layer. We have added the information in the revised manuscript. (page 8, lines 21-23)

For typhoon Haiyan, we have carried out several tests with and without the nudging options on the 3 km resolution grid. The 3 km resolution grid is for our control experiment in this study. The tests revealed that, simulations without nudging during the '24-h pre-sensitivity stage' produced larger typhoon track errors than that with a nudging stage. Regarding the TC structure and intensity, we did not find significant differences in the Haiyan case between the experiments with and without the nudging

treatment. We did not perform any nudging test on the 1 km resolution grid, because of our limited computational resource.

4. *P. 8, lines 5-10: It is not clear to the reader why 1 km F2 test is omitted. The authors indicate that the "the simulation result of F2 is somewhat between those of F0 and F1 for other resolutions, we omitted the F2 test at 1-km resolution". Can you extrapolate that for the 1 km resolution? I think the authors should elaborate on F2 at 1 km.*

We have carried out the experiment of 1 km F2 as suggested. This added experiment was also suggested in comment 1 by referee 1. Accordingly, we have modified the following figures for adding the experiments with flux option F2: Figs. 4, 6, 8, 9, 10, 11, 12, 13, 14, and 15. We have also modified some related sentences in the revised manuscript. The majority of them can be found in sub-section 2.3 (P8, Experimental designs), section 4 (P12, P14, P15, P16), section 5 (P18). Please find the respective changes in the revised manuscript.

With the new added 1 km F2, the observed typhoon intensity is located between 1 km F2 and 1 km F1 experiments. Overall, the simulated typhoon intensity (and other features presented) of F2 is again somewhat between those of F0 and F1 at the resolution of 1 km.

5. *I cannot find the observed structure of the TC in the manuscript, which would be very important for building confidence on the modelling results.*

We do not have observations for the structure of typhoon Haiyan. In the former manuscript, we have provided a reference (Shimada et al. 2018) which revealed the observational information for the structure of Haiyan (2013). Please find the information on page 11 (lines 30-34) and page 12 (lines 1-5).

References

Bryan G. H., Wyngaard, J. C., and Fritsch, J. M.: Resolution requirements for the simulation of deep moist convection, Mon. Wea. Rev., 131, 2394–2416, 2003.

Davis C., and Coauthors: Prediction of landfalling hurricanes with the Advanced Hurricane WRF model, Mon. Wea. Rev.,136, 1990–2005, doi:10.1175/2007MWR2085.1, 2008.

Deng, A. and Stauffer, D. R.: On improving 4-km mesoscale model simulations, J. Appl. Meteor., 45, 361-381, 2006.

Gentry M. S. and Lackmann, G. M.: Sensitivity of simulated tropical cyclone structure and intensity to horizontal resolution, Mon. Wea. Rev., 138, 688–704, doi:10.1175/2009MWR2976.1, 2010.

Kotroni, V. and Lagouvardos, K.: Evaluation of MM5 High-Resolution Real-Time Forecasts over the Urban Area of Athens, Greece, Journal of Applied Meteorology, 43, 1666-1678, doi:10.1175/JAM2170.1, 2004.

Lee, S.-W., Lee, D.-K., and Chang, D.-E.: Impact of horizontal resolution and cumulus parameterization scheme on the simulation of heavy rainfall events over the Korean Peninsula, Advances in Atmospheric Sciences 28, 1-15, doi:10.1007/s00376-010-9217-x, 2011.

Li, F., Song, J., and Li, X.: A preliminary evaluation of the necessity of using a cumulus parameterization scheme in high-resolution simulations of Typhoon Haiyan (2013), Natural Hazards, 92, 647-671, doi:10.1007/s11069-018-3218-y, 2018.

Miyamoto Y., Kajikawa, Y., Yoshida, R., Yamaura, T., Yashiro, H., and Tomita, H.: Deep moist atmospheric convection in a subkilometer global simulation, Geophy. Res. Lett., 40, 4922–4926, doi:10.1002/grl.50944, 2013.

On, N., Kim, H. M., and Kim, S.: Effects of Resolution, Cumulus Parameterization Scheme, and Probability Forecasting on Precipitation Forecasts in a High-Resolution Limited-Area Ensemble Prediction System, Asia-Pacific Journal of Atmospheric Sciences 54, 623-637, doi:10.1007/s13143-018-0081-4, 2018.

Shimada U., Kubota, H., Yamada, H., Cayanan, E.O., and Hilario, F. D.: Intensity and Inner-Core Structure 5 of Typhoon Haiyan (2013) near Landfall: Doppler Radar Analysis, Mon. Wea. Rev., 146, 583–597, doi:10.1175/MWR-D-17-0120.1,2018.

Sun, Y., Yi, L., Zhong, Z., Hu, Y., and Ha, Y.:, Dependence of model convergence on horizontal resolution and convective parameterization in simulations of a tropical cyclone at gray-zone resolutions, J. Geophys. Res. Atmos., 118,7715–7732, doi:10.1002/jgrd.50606, 2013.

Warner, T. T. and Hsu, H. M.: Nested-model simulation of moist convection: the impact of coarse-grid parameterized convection on fine-grid resolved convection, Mon. Wea. Rev., 128, 2211–2231, 2000.

Weisman, M. L., Skamarock, W. C., and Klemp, J. B.: The resolution dependence of explicitly modeled convective systems, Mon. Wea. Rev., 125, 527–548, 1997.

Yu, X. and Lee, T.-Y.: Role of convective parameterization in simulations of heavy precipitation systems at grey-zone resolutions — case studies, Asia-Pacific Journal of Atmospheric Sciences 47, 99-112, doi:10.1007/s13143-011-0001-3, 2011.

---

## Author Comment (AC6)

*Supplement of*

**Effects of Horizontal Resolution and Air–Sea Flux Parameterization on the Intensity and Structure of simulated Typhoon Haiyan (2013)**

Kueh et al.

This supplement contains 2 figures to show the tracks and central positions of the simulated typhoon Haiyan during the mature stage.

[Figure]

Fig. S1. The central positions of simulated Haiyan during the time period from 1200UTC 7 November 2013 to 0000UTC 8 November 2013. The dots in tan denote the WMO best track positions.

[Figure]

Fig. S2. The simulated tracks and central positions within a 5-hour time slot. For each experiment, the central time of the 5-hour time slot for the hourly positions is taken as the simulated typhoon centered at its nearest location upon the best-track location at 1800UTC 7 November 2013. The simulated position at each central time of the respective 5-hour time slot is denoted by a larger symbol. The dot in tan denotes the WMO best track positions at 1800UTC 7 November 2013.

---

## Author Response (AR1)

**Effects of Horizontal Resolution and Air–Sea Flux Parameterization on the Intensity and Structure of simulated Typhoon Haiyan (2013)**

Mien-Tze Kueh, Wen-Mei Chen, Yang-Fan Sheng, Simon C. Lin, Tso-Ren Wu, Eric Yen, Yu-Lin Tsai, Chuan-Yao Lin

**Response to referee 1**

Dear Referee,

We thank the referee for the comments and suggestions to our manuscript. Please find our responses to your comments and questions below. The comments and questions are given in italics, and our responses are in blue.

*GENERAL COMMENTS:*

*Through numerical simulation experiments and analysis, the authors present a study aimed at evaluating the influence of horizontal resolution and surface flux formulas on the development of Typhoon. The results show the increase of resolution and more reasonable surface flux formulas leads to an improvement of the typhoon intensity simulation. Although the innovation of the paper is not strong, the relevant conclusions of the paper have a positive impact on typhoon forecast.*
*It is clear a lot of effort has gone into this research. However, I question the design of the sensitivity experiments. As such, the recommendation is for major revisions, and this reviewer feels this manuscript will be improved by addressing the comments below.*

We thank the referee's positive comments to our manuscript. We have carried out the numerical experiment of 1 km F2 in response to your specific comment 1. The revised manuscript has been edited by *Wallace Academic Editing* and is considered to be improved in grammar, punctuation, general readability, and native English usage. In addition, we followed the suggestion of *Wallace Academic Editing* and reworded the "impacts" to "effects" in our title, which is now:

> Effects of Horizontal Resolution and Air–Sea Flux Parameterization on the Intensity and Structure of simulated Typhoon Haiyan (2013)

Below are our point-by-point replies to your comments.

*SPECIFIC COMMENTS*

1. *On P8 line 5-9, "Because the simulation result of F2 is somewhat between those of F0 and F1 for other resolutions, we omitted the F2 test at 1-km resolution." However, judging from the time evolution of typhoon intensity, the observed typhoon intensity is located between 1km F0 and 1km F1 experiments. Is there a better coincidence between 1km F2 and observation?*
*On P7 line 20-25, the author also mentioned that "F1 predicts larger CH and CQ at all wind speeds than F2 does, implying that F1 has a potentiality to gain larger enthalpy fluxes", This may also be the reason why the simulated typhoon intensity of 1km F1 is higher than that observed.*
*Overall, I think the numerical test of 1 km F2 is very critical and cannot be omitted.*

We have carried out the experiment of 1 km F2 as suggested. This added experiment was also suggested in comment 4 by referee 2. Accordingly, we have modified the following figures for adding the experiments with flux option F2: Figs. 4, 6, 8, 9, 10, 11, 12, 13, 14, and 15. We have also modified some related sentences in the revised manuscript. The majority of them can be found in sub-section 2.3 (P8, Experimental designs), section 4 (P12, P14, P15, P16), section 5 (P18). Please find the respective changes in the revised manuscript.

With the new added 1 km F2, the observed typhoon intensity is now located between 1 km F2 and 1 km F1 experiments. The simulated typhoon intensity in 1 km F2 is relatively closer to that in 1 km F0, not 1km F1. We can't say there is a better coincidence between 1 km F2 and observation. Although the experiment 1 km F1 overpredicts the typhoon intensity, overall it still produces the best solution among others.

Regarding the statement quoted here, "*F1 predicts larger CH and CQ at all wind speeds than F2 does, implying that F1 has a potentiality to gain larger enthalpy fluxes*", yes, we consider this to be the reason for the most intense typhoon found in 1 km F1 among others in the 1 km group.

2. *On P4 line 5, the horizontal grid spacing is 1, 3 and 6 km. So why do the authors still use the updated Kain-Fritsch convective scheme? If the convective scheme is not used, do the simulation results change?*

The convection parameterization (K-F cumulus scheme) was used for all resolutions. This is because we intend to keep consistency among all cases. The convective treatment was also mentioned in the comment 1 by referee 2. For typhoon Haiyan, we have carried out several tests with and without cumulus parameterization on the 3 km resolution grid. The 3 km resolution grid is for our control experiment in this study. The tests revealed that, simulations with and without cumulus parameterization produced overall similar simulated storm intensity. We did not perform any

convection parameterization test on the 1 km resolution grid, because of our limited computational resource.

Numerous studies have suggested that 3-4 km resolutions without any cumulus parameterization is sufficient to represent mesoscale convections (e.g.,Weisman et al. 1997; Davis et al. 2008; Gentry and Lackmann, 2010). However, such a grid resolution is still insufficient for representing individual convective cells (e.g., Bryan et al. 2003; Miyamoto et al. 2013). The use of cumulus parameterization with the grid spacing below 3-4 km has been investigated by a number of recent studies. Some studies suggested to activate the cumulus parameterization for simulation of moist convective event with a grid resolution of 4 km (Deng and Stauffer 2006), 3 km (Lee et al 2011) and 2 km (Kotroni and Lagouvardos 2004). Some others, however, revealed that the activation of cumulus parameterization for simulation with grid spacings of 2-3 km produced overall similar simulated storm as in the simulation with explicit convection (e.g., Yu et al. 2011; Li et al. 2018; On et al. 2018). Sun et al. (2013) studied the appropriateness of a variety of cumulus parameterization schemes used in high-resolution simulations. They assumed that the cumulus scheme is closely related to the model convergence in simulating TC intensity. Here, a convergence of model solution in terms of TC intensity is that the simulated TC intensity would remain similar irrespective of any further reduction of the grid spacing. They found a weak convergence in fine resolution (from 3 to 1 km) simulations with most of the schemes, whereas the convergence is relatively strong in the simulations with a scale-aware scheme designed for any resolution. Accordingly, cumulus parameterization may still play a role in the fine resolution (3 to 1 km) simulations. The question then arises as to what is the appropriate design of cumulus parameterization for very high resolution. However, this is far beyond the scope of our present study.

3. *On P8 line 10, the general simulation usually chooses to gradually increase the resolution through nesting, while all resolution simulations in this paper do not use nesting. Why?*

Our large single domain was chosen to cover the majority of simulated Haiyan (2013)'s convection during the period of sensitivity simulation, and to make a cleaner comparison among those experiments running at different resolutions.

Higher-resolution nested model configuration are widely used in numerical weather prediction and regional climate modelling. The main reason for this is because large area of high-resolution model simulation is computationally too expensive. However, consistency between nested grids is also important. With lateral boundaries on multiple grids, model solutions may not be smooth across nested-domain boundaries. In a nested WRF simulation, a discontinuity in precipitation and moisture fields (i.e., a sharp gradient) across the inner domain boundaries has long been recognized by the WRF community. Uncertainty related to the use of multiple nested grids can resulted from mismatched model physics across nested-domain boundaries. For example, Warner and Hsu (2000) revealed that the treatment of convection on the outer grid can affect the explicit convection on the inner grid. Their

result indicated that the simulation biases related to the parameterized convection (e.g., errors in precipitation timing, precipitation intensity, and the vertical distribution of latent heating) can greatly modulate the explicit convection on the inner grid through the induced subsidence from the outer grid.

4. *Section 2 should be divided into subsections, such as 2.1 moisture roughness length, 2.2 momentum roughness length, 2.3 bulk transfer coefficients, 2.4 experimental designs.*

We have followed the suggestion of the referee. In the revised manuscript, section 2 has been divided into 3 subsections:
2.1 Flux parameterizations in WRF-ARW (page 4)
2.1.1 Momentum and moisture roughness length for flux option F0 (page 4)
2.1.2 Momentum and moisture roughness length for flux option F1 (page 5)
2.1.3 Momentum and moisture roughness length for flux option F2 (page 6)
2.2 Comparison between modelled and observational bulk transfer coefficients (page 7)
2.3 Experimental designs (page 8)
Please find the respective changes in the revised manuscript.

5. *On P8 line16-17, please show the original resolution of GFS data used.*

The GFS data has a horizontal resolution of 0.5 degree. We have added this information to the revised manuscript (P8, lines 25-26).

6. *From Fig. 4a, it is difficult to distinguish the track difference between different sensitivity tests, and it is suggested to modify this figure.*

We have decided to not modify the plot, because it is still hard to distinguish the track difference between different sensitivity tests even when different symbols were used to denote them. We changed the layout of Fig. 4 instead, so as to make the track plot larger. In addition, we added 2 supplemental figures (Figs. S1 and S2) to show the tracks and central positions of the simulated storms during the mature stage. We prefer to put the detailed information of tracks/positions in a supplement of the manuscript, because we feel that this is not absolutely necessary in the context of our present study. In the revised manuscript, we mentioned the 2 supplemental figures in the figure caption of Fig.4.

The figures are also shown below for your reference:

In Fig. S1, we show the central positions of simulated Haiyan during the time period from 1200UTC 7 November 2013 to 0000UTC 8 November 2013. The simulated tracks and central positions within a 5-hour time slot are shown in Fig. S2. For each experiment, the central time of the 5-hour time slot for

picking the hourly positions is taken as the simulated typhoon centered at its nearest location upon the best-track location at 1800UTC 7 November 2013.

[Figure]

Fig. S1

[Figure]

Fig. S2

7. *On P9 Line 6, "The intensity is not very sensitive to the resolutions of 3 and 6 km…", while I think it is sensitive enough.*

We agree. In the revised manuscript, this sentence has been modified (P9, lines 11-13):
'The intensity is sensitive to model resolution. Overall, the intensity increases as the resolution is changed from 6 to 3 km, but it significantly increases as grid spacing is reduced to 1 km.'

8. *There is no observed typhoon structure in this article as a comparison, and it is suggested to add corresponding figures.*

We do not have observations for the structure of typhoon Haiyan. In the former manuscript, we have provided a reference (Shimada et al. 2018) which revealed the observational information for the structure of Haiyan (2013). Please find the information on page 11 (lines 30-34) and page 12 (lines 1-5).

9. *The authors can introduce the use of contoured frequency by altitude diagrams. Maybe some readers are not familiar to those figures.*

We have added a paragraph to briefly introduce the use of CFADs. In the revised manuscript, these sentences have been added (from P16 lines 27-32 to P17 lines 1-7):

The CFADs is a statistical method for summarizing the vertical distributions of meteorological fields. A CFAD is constructed by collecting frequency distributions of a particular variable at evenly spaced altitudes within an area, compiling them into a two-dimensional (data bin and altitude) data set, and portraying the data on a single contour plot. The ordinate in the plot represents the altitude variation, and the abscissa the frequency bin. For each altitude on a CFAD, the frequencies should add up to 100 %. For a given CFAD, each point depicts the frequency of occurrence of the data in that bin at a specific altitude. Accordingly, the CFADs ignore horizontal variability and provide a bulk statistical measure for comparing the vertical structure of evolving fields of cumulonimbus clouds or any convective systems. Here, we take Fig. 13a as an example. On this CFAD, there are higher percentages (e.g., bounded by 20 %) found in the higher reflectivity bins at lower altitudes and found in the lower reflectivity bins at higher altitudes. The former can indicate convective cells or precipitations, whereas the latter can indicate snow or stratiform precipitation. Therefore, consider a set of CFADs for an evolving cumulonimbus clouds, say from the initiation to the mature stage, the maximum percentage at each altitude would change from vertically oriented to negatively tilted toward lower reflectivity values. More detailed interpretations of the use of CFADs can be found in the work by Yuter and Houze (1995).

10. *In general, this quality of this paper is a little difficult to be understood by readers. I hope the authors can carefully revise some long sentences to make it easy to understand.*

The revised manuscript has been edited by *Wallace Academic Editing* and is considered to be improved in grammar, punctuation, general readability, and native English usage.

11. *Why doesn't the positive effect of reasonable surface flux formulas be enhanced efficiently if the grid spacing is relatively large? It should explain more clearly in the paper.*

The surface fields and vertical structures between different resolution groups are more significant than that between flux options (cf. Figs. 6,8,9,10, and 11). The analyses indicate that the typhoon intensity is mainly controlled by the model resolution. Therefore, there are higher frequency of occurrences for extremely high wind speeds found in the experiments with higher resolution. From the description of the flux options given in section 2, we understand that near-surface wind speed is an influential factor for the calculation of bulk transfer coefficients. Accordingly, it is because of the low frequency of occurrences for extremely high wind speeds, the positive effect of the more reasonable surface option (F1) cannot be enhanced efficiently with a relatively large grid spacing of 6 km.

We have added several sentences to explain the above concept in the revised manuscript. The referee can find them on pages 13 (lines 29-32), 16 (lines 10-17), and 20 (lines 3-6).

*TECHNICAL CORRECTIONS*

We are very grateful for your careful corrections. In the revised manuscript, we have amended these sentences as suggested.

*1. Page 3, Line 27, change "reaches" to "reached"*   Done. Now on page 3, line 24.
*2. Page 7, Line 29, change "were" to "was"*
Now on page 8, line 4. This should be a plural verb.
*3. Page 7, Line 31, change "level-off" to "level-offs"* Done. Now on page 8, line 5.
*4. Page 8, Line 3, change "between" to "among"*     Done. Now on page 8, line 10.
*5. Page 8, Line 4, this sentence is incomplete.*
We have amended this sentence: "Accordingly, F1 is expected to have the highest potential to achieve the most intense storm among the three options because *it has* the highest values of $C_K/C_D$ almost at all wind speeds." Now on page 8, line 9-10.
*6. Page 9, Line 30, change "agree" to "agrees"*                Done. Now on page 10, line 5.

*7. Page 11, Line 18, change "enhanced" to "enhancement"*    Done. Now on page 11, line 24.
*8. Page 12, Line 29, change "increases" to "increase"*    Done. Now on page 13, line 1.
*9. Page 13, Line 22, change "detail" to "detailed"*    Done. Now on page 13, line 33.
*10. Page 13, Line 23, change "increase" to "increases"*    Done. Now on page 13, line 34.
*11. Page 15, Line 5, change "between" to "among"*    Done. Now on page 15, line 16.
*12. Page 15, Line 13, change "consider" to "considering"*    Done. Now on page 15, line 25.
*13. Page 15, Line 24, change "that" to "than"*    Done. Now on page 16, line 1.
*14. Page 17, Line 26, change "represent" to "represents"*    Done. Now on page 18, line 21.
*15. Page 18, Line 21, change "fluxes" to "flux"*    Done. Now on page 19, line 16.
*16. Page 18, Line 32, change "later" to "layer"*    Done. Now on page 19, line 27.
*17. Page 19, Line 19, change "as" to "for"*    Done. Now on page 20, line 16.

**Response to referee 2**

Dear Referee,

We thank the referee for the comments and suggestions to our manuscript. Please find our responses to your comments and questions below. The comments and questions are given in italics, and our responses are in blue.

*This contribution aims at investigating the impacts of model horizontal resolution and surface flux formulas on structure and intensity of tropical cyclones. The authors intend to study the sensitivity of the intensity of tropical cyclones to surface fluxes parameterizations of the Weather Research and Forecasting Model (WRF) at very high resolutions (1, 3 and 6 km, when model resolution approaches the convective scale).*
*The paper is in general structured and scientifically sound. English should be revised by a native speaker in order to correct some typos, but also to rephrase long sentences that make the manuscript hard to read. Scientifically, there are several limitations, especially regarding the design of the simulations, that require major revisions of the submitted version.*

We thank the referee's positive comments to our manuscript. We have carried out the numerical experiment of 1 km F2 in response to your comment 4. The revised manuscript has been edited by *Wallace Academic Editing* and is considered to be improved in grammar, punctuation, general readability, and native English usage. In addition, we followed the suggestion of *Wallace Academic Editing* and reworded the "impacts" to "effects" in our title, which is now:

> Effects of Horizontal Resolution and Air–Sea Flux Parameterization on
> the Intensity and Structure of simulated Typhoon Haiyan (2013)

Below are our point-by-point replies to your comments.

1. *Is convection parameterized (K-F cumulus scheme) for all resolutions? 6 km simulations are in the grey-zone, but cumulus should be explicitly resolved when working at resolutions below 3-4 km. How this parameterization impacts the results?*

Yes, the convection parameterization (K-F cumulus scheme) was used for all resolutions. This is because we intend to keep consistency among all cases. The convective treatment was also mentioned in the comment 2 by referee 1. For typhoon Haiyan, we have carried out several tests with and without cumulus parameterization on the 3 km resolution grid. The 3 km resolution grid is for our control experiment in this study. The tests revealed that, simulations with and without cumulus parameterization produced overall similar simulated storm intensity. We did not perform any convection parameterization test on the 1-km resolution grid, because of our limited computational resource.

Numerous studies have suggested that 3-4 km resolutions without any cumulus parameterization is sufficient to represent mesoscale convections (e.g.,Weisman et al. 1997; Davis et al. 2008; Gentry and Lackmann, 2010). However, such a grid resolution is still insufficient for representing individual convective cells (e.g., Bryan et al. 2003; Miyamoto et al. 2013). The use of cumulus parameterization with the grid spacing below 3-4 km has been investigated by a number of recent studies. Some studies suggested to activate the cumulus parameterization for simulation of moist convective event with a grid resolution of 4 km (Deng and Stauffer 2006), 3 km (Lee et al 2011) and 2 km (Kotroni and Lagouvardos 2004). Some others, however, revealed that the activation of cumulus parameterization for simulation with grid spacings of 2-3 km produced overall similar simulated storm as in the simulation with explicit convection (e.g., Yu et al. 2011; Li et al. 2018; On et al. 2018). Sun et al. (2013) studied the appropriateness of a variety of cumulus parameterization schemes used in high-resolution simulations. They assumed that the cumulus scheme is closely related to the model convergence in simulating TC intensity. Here, a convergence of model solution in terms of TC intensity is that the simulated TC intensity would remain similar irrespective of any further reduction of the grid spacing. They found a weak convergence in fine resolution (from 3 to 1 km) simulations with most of the schemes, whereas the convergence is relatively strong in the simulations with a scale-aware scheme designed for any resolution. Accordingly, cumulus parameterization may still play a role in the fine resolution (3 to 1 km) simulations. The question then arises as to what is the appropriate design of cumulus parameterization for very high resolution. However, this is far beyond the scope of our present study.

2. *P. 4:  What is the nesting approach followed in the simulations? Is it a one way or a two way nesting?*

No nesting approach was used for our simulations in this study. Our large single domain was chosen to cover the majority of simulated Haiyan (2013)'s convection during the period of sensitivity simulation, and to make a cleaner comparison among those experiments running at different resolutions.

Higher-resolution nested model configuration are widely used in numerical weather prediction and regional climate modelling. The main reason for this is because large area of high-resolution model simulation is computationally too expensive. However, consistency between nested grids is also important. With lateral boundaries on multiple grids, model solutions may not be smooth across nested-domain boundaries. In a nested WRF simulation, a discontinuity in precipitation and moisture fields (i.e., a sharp gradient) across the inner domain boundaries has long been recognized by the WRF community. Uncertainty related to the use of multiple nested grids can resulted from mismatched model physics across nested-domain boundaries. For example, Warner and Hsu (2000) revealed that the treatment of convection on the outer grid can affect the explicit convection on the inner grid. Their result indicated that the simulation biases related to the parameterized convection (e.g., errors in precipitation timing, precipitation intensity, and the vertical distribution of latent heating) can greatly modulate the explicit convection on the inner grid through the induced subsidence from the outer grid. The nesting issue was also mentioned in the comment 3 by referee 1.

3. *Nudging is critical for correctly representing the TC structures, their paths and intensities. So, ideally, some sensitivity runs should take nudging options into account. If not possible, further details about nudging approach followed and its impacts on the results should be elaborated.*

In this study, the simulations were performed with two stages. The nudging stage is a 24-hour period before the 78 hours sensitivity stage. No analysis nudging was applied to the model grid during the entire period of sensitivity experiment.

In this study, the analysis nudging was applied to the horizontal wind components, potential temperature, and water vapor mixing ratio. The nudging coefficients for all variables were set at 0.0003 $s^{-1}$. The nudging was only applied at all levels above the planetary boundary layer. We have added the information in the revised manuscript. (page 8, lines 21-23)

For typhoon Haiyan, we have carried out several tests with and without the nudging options on the 3 km resolution grid. The 3 km resolution grid is for our control experiment in this study. The tests revealed that, simulations without nudging during the '24-h pre-sensitivity stage' produced larger typhoon track errors than that with a nudging stage. Regarding the TC structure and intensity, we did not find significant differences in the Haiyan case between the experiments with and without the nudging

treatment. We did not perform any nudging test on the 1 km resolution grid, because of our limited computational resource.

4. *P. 8, lines 5-10: It is not clear to the reader why 1 km F2 test is omitted. The authors indicate that the "the simulation result of F2 is somewhat between those of F0 and F1 for other resolutions, we omitted the F2 test at 1-km resolution". Can you extrapolate that for the 1 km resolution? I think the authors should elaborate on F2 at 1 km.*

We have carried out the experiment of 1 km F2 as suggested. This added experiment was also suggested in comment 1 by referee 1. Accordingly, we have modified the following figures for adding the experiments with flux option F2: Figs. 4, 6, 8, 9, 10, 11, 12, 13, 14, and 15. We have also modified some related sentences in the revised manuscript. The majority of them can be found in sub-section 2.3 (P8, Experimental designs), section 4 (P12, P14, P15, P16), section 5 (P18). Please find the respective changes in the revised manuscript.

With the new added 1 km F2, the observed typhoon intensity is located between 1 km F2 and 1 km F1 experiments. Overall, the simulated typhoon intensity (and other features presented) of F2 is again somewhat between those of F0 and F1 at the resolution of 1 km.

5. *I cannot find the observed structure of the TC in the manuscript, which would be very important for building confidence on the modelling results.*

We do not have observations for the structure of typhoon Haiyan. In the former manuscript, we have provided a reference (Shimada et al. 2018) which revealed the observational information for the structure of Haiyan (2013). Please find the information on page 11 (lines 30-34) and page 12 (lines 1-5).

[revised manuscript text omitted]

已删除: for both F0 and F1

已删除: the

已删除: st

已删除: compared with

已删除: figures

已删除: in

已删除: and stronger

已删除: dilution

已删除: the

已删除: mixing

已删除: To a large degree

已删除: . Skamarock (2004)

已删除: around

已删除: represent

已删除: here

已删除: about

已删除: compared with

已删除: , namely

已删除: very

已删除: However, Bryan et al. (2003) have

of 1 km remains insufficient for explicitly resolving convective clouds. Here, we speculate that insufficient intense convection cores in our simulations may result in the absence of peak values around the respective model effective resolutions.

Finally, the slope of the physical portion of the spectra (for the horizontal and vertical components, respectively) remains essentially unchanged as the model resolution is varied. In general, the higher the resolution, the further is the downscale extent, the smaller are the scales represented, and the smaller are model effective resolutions reached. The aforementioned descriptions remain true for all flux options and averaging layers in our work. The key point is that the effective resolution is determined by grid spacing, not the flux options. We noted some subtle differences between the F0 and F1 experiments at smaller wavelengths at nearly 2 DX in the KE spectra, but in only a few hours of the entire simulation period. In short, although the use of more reasonable flux formulas can increase simulated storm intensity to some extent, the positive effect of the flux formulas cannot be efficiently enhanced unless the grid spacing is appropriately reduced to yield intense and contracted eyewall structure.

**6 Conclusions**

This study investigated the effects of resolution and surface flux formulas on typhoon intensity and structure simulations through the case study of Super Typhoon Haiyan (2013). The effect could be separated between horizontal resolution and air–sea flux parameterization through the inner-core structure and spectral analyses. Specifically, we found significantly increased sensitivity of TC intensity simulations to surface flux parameterizations when model resolution approached the convective scale (~ 1 km).

Three sets of surface flux formulas in the WRF model were tested using grid spacings of 1, 3, and 6 km. Increased resolution and more reasonable flux parameterization could both improve typhoon intensity simulation to a certain extent, but their effects on storm structures differed. A combination of decreasing momentum transfer coefficient and increasing enthalpy transfer coefficients tends to yield stronger storm. The storm size (in terms of the radius of maximum winds) is apparently sensitive to the changes in grid spacing. The choice of flux formulas had little effect on storm size. Sufficiently high resolution was more conducive to the positive effect of flux formulas on simulated typhoon intensity.

Reducing the grid spacing to 1 km yielded a deeper, stronger, and more upright and contracted eyewall. Stronger inflow and outflow as well as relatively wider and weaker eyewall updraft were found in the lower resolutions, indicating a stronger secondary circulation. As strength increased, the eyewall contracted, indicating higher efficiency of storm intensification in smaller grid spacing. This contraction of eyewall is associated with an upper-level warming layer, which signifies an intense development of the simulated typhoon. The upper-level warming process can be attributed to the air detrained from the intense updraft cores in the eyewall, which can only be resolved by higher resolutions, such as 1 or 3 km.

The analysis of the CFADs revealed that the size of updraft cores in the eyewall shrinks and the region of downdraft increases, and both updraft and downdraft become more intense as the resolution increases. Therefore, the intense reflectivity (convective) cores of the higher resolution simulations are driven by more intense updrafts within a rather small fraction of the spatial area,

已删除: Therefore
已删除: we suggest that
已删除:
已删除: impacts
已删除: the
已删除: In this study, the
已删除: fluxes
已删除: s
已删除: different
已删除: Both i
已删除: impacts
已删除: were different
已删除: impact
已删除: later
已删除: is increased

[revised manuscript text omitted]

---

## Author Response (AR2)

**Effects of Horizontal Resolution and Air–Sea Flux Parameterization on the Intensity and Structure of simulated Typhoon Haiyan (2013)**

Mien-Tze Kueh, Wen-Mei Chen, Yang-Fan Sheng, Simon C. Lin, Tso-Ren Wu, Eric Yen, Yu-Lin Tsai, Chuan-Yao Lin

**Response to referee 1**

Dear Referee,

We have performed an additional experiment for providing more evidence about the impact of the cumulus parameterization on the simulation with 1 km resolution. Please find our responses to your questions and comments below. The questions and comments are given in italics, and our responses are in blue.

*GENERAL COMMENTS:*
*In the reply, the author claimed that the simulations with and without cumulus parameterization produced overall similar simulated storm intensity. Is it means that the cumulus parameterization may not play a role in the fine resolution (3 km) simulation? The authors should provide more evidence about the impact of the cumulus parameterization on the simulation in this case with 1 km resolution. Why must the cumulus parameterization be used in the 1 km resolution? I hope the author could perform the experiments in 1km resolution with and without cumulus parameterization and show the differences between the two tests.*

We have carried out an additional experiment of 1 km resolution without cumulus parameterization in response to your specific comment. The additional experiment used the same configuration as 1kmF0, but with the cumulus parameterization turned off, hereafter 1kmF0_noCU. The result indicates that the use of a cumulus parameterization in such a high resolution (1 km) simulation does not exert considerable influence on the typhoon intensity, track, and structure. For this additional experiment, we have conducted the same analysis as that for those experiments presented in this manuscript. There are very little difference between 1kmF0 and 1kmF0_noCU in terms of the minimum central pressure and track (cf. Fig 4), surface fields (cf. Figs. 6, 8, and 9), vertical structure (cf. Figs. 10 and 11), upper layer warming (cf. Fig 12), CFADs for simulated reflectivity and vertical motion (cf. Figs. 13, 14, 15), and also the KE spectra (cf. Fig. 16). Specifically, we compared the result of 1kmF0_noCU to that of

1kmF0, 1kmF2 (and also 1kmF1), and 3kmF0. The comparison reveals that, results of 1kmF2 (and 1kmF1) and 3kmF0 are apparently different from 1kmF0_noCU (so as 1kmF0). This is to say, the use of different flux options (1kmF1 & 1kmF2) and spatial resolution (3kmF0) can exert large impact on the simulation of typhoon Haiyan, but the cumulus parameterization does not play a role in such a fine resolution (1 km) simulation. Therefore, in this manuscript, we used the cumulus parameterization for all resolution to keep consistency among all cases.

In this revision, we added a supplemental figure to show the simulated minimum central pressures and tracks of 1kmF0_noCU and 1kmF0. We did not put all the other plots for 1kmF0_noCU therein because we do not want to overemphasize the use of cumulus parameterization in this article.

In page 9, lines 17-25, we added the following sentences:

*In this study, the cumulus parameterization was used for all resolutions. This is because we intend to keep consistency among all cases. Results from an additional simulation of the 1kmF0 with the cumulus parameterization turned off revealed that the simulated Haiyan intensity and structure are overall similar. The minimum central pressure and storm track of the 1kmF0 without cumulus parameterization follow almost the same evolutions as that of 1kmF0 (supplemental figure Fig. S3). In our case, the use of a cumulus parameterization in high resolution (1 km) simulation does not exert considerable influence on the typhoon intensity, track, and structure. Some studies have also revealed that the activation of cumulus parameterization for simulation with grid spacings of 2-3 km produced overall similar simulated storm as in the simulation with explicit convection (e.g., Yu et al. 2011; Li et al. 2018; On et al. 2018).*

Below is the supplemental figure for your reference:

[Figure]

Fig. S3. The simulated minimum central pressures (hPa) and tracks during the period from 0000UTC 5 November 2013 through 0600UTC 8 November 2013. The experiment 1kmF0 is denoted in blue, and the additional experiment for simulation with 1 km resolution (and F0) but without cumulus parameterization is denoted in red. The evolutions of the simulated minimum central pressures (top panel) are shown in relation to longitude. The simulated tracks (bottom panel) were derived from the 1-hourly simulation results. The simulated position at each 0000UTC is denoted by a dot.